# TOWARDS MARGINAL FAIRNESS SLICED WASSERSTEIN BARYCENTER

**Khai Nguyen**[*]
Department of Statistics and Data Sciences
University of Texas at Austin
Austin, TX 78713, USA
`khainb@utexas.edu`

**Hai Nguyen**[*]
Qualcomm AI Research[†]
`hainn@qti.qualcomm.com`

**Nhat Ho**
Department of Statistics and Data Sciences
University of Texas at Austin
Austin, TX 78713, USA
`minhnhat@utexas.edu`

## ABSTRACT

The Sliced Wasserstein barycenter (SWB) is a widely acknowledged method for efficiently generalizing the averaging operation within probability measure spaces. However, achieving marginal fairness SWB, ensuring approximately equal distances from the barycenter to marginals, remains unexplored. The uniform weighted SWB is not necessarily the optimal choice to obtain the desired marginal fairness barycenter due to the heterogeneous structure of marginals and the non-optimality of the optimization. As the first attempt to tackle the problem, we define the marginal fairness sliced Wasserstein barycenter (MFSWB) as a constrained SWB problem. Due to the computational disadvantages of the formal definition, we propose two hyperparameter-free and computationally tractable surrogate MFSWB problems that implicitly minimize the distances to marginals and encourage marginal fairness at the same time. To further improve the efficiency, we perform slicing distribution selection and obtain the third surrogate definition by introducing a new slicing distribution that focuses more on marginally unfair projecting directions. We discuss the relationship of the three proposed problems and their relationship to sliced multi-marginal Wasserstein distance. Finally, we conduct experiments on finding 3D point-clouds averaging, color harmonization, and training of sliced Wasserstein autoencoder with class-fairness representation to show the favorable performance of the proposed surrogate MFSWB problems[1].

## 1 INTRODUCTION

Wasserstein barycenter (Agueh & Carlier, 2011) generalizes "averaging" to the space of probability measures. In particular, a Wasserstein barycenter is a probability measure that minimizes a weighted sum of Wasserstein distances between it and some given marginal probability measures. Due to the rich geometry of the Wasserstein distance (Peyré & Cuturi, 2020), the Wasserstein barycenter can be seen as the Fréchet mean (Grove & Karcher, 1973) on the space of probability measures. As a result, Wasserstein barycenter has been applied widely to various applications in machine learning such as Bayesian inference (Srivastava et al., 2018; Staib et al., 2017), domain adaptation (Montesuma & Mboula, 2021), clustering (Ho et al., 2017), sensor fusion (Elvander et al., 2018), text classification (Kusner et al., 2015), and so on. Moreover, Wasserstein barycenter is also a powerful tool for computer graphics since it can be used for texture mixing (Rabin et al., 2012), style transfer (Mroueh, 2020), shape interpolation (Solomon et al., 2015), and many other tasks on many other domains.

---

[*]Equal Contribution
[†]Qualcomm Vietnam Company Limited
[1]Code for the paper is published at `https://github.com/khainb/MFSWB`.

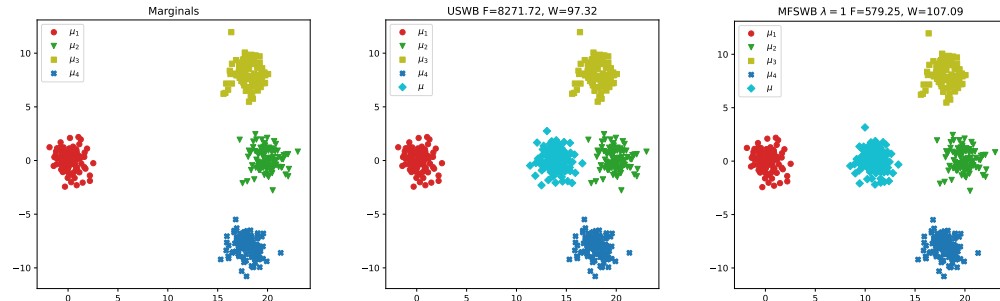

Figure 1: The uniform SWB and the MFSWB of 4 Gaussian distributions.

Despite being useful, it is very computationally expensive to compute Wasserstein barycenter. In more detail, the computational complexity of Wasserstein barycenter is $\mathcal{O}(n^3 \log n)$ when using linear programming (Anderes et al., 2016) where $n$ is the largest number of supports of marginal probability measures. When using entropic regularization for optimal transport (Cuturi, 2013), the computational complexity is reduced to $\mathcal{O}(n^2)$ (Kroshnin et al., 2019). Nevertheless, quadratic scaling is not enough when the number of supports approaches a hundred thousand or a million. To address the issue, Sliced Wassserstein Barycenter (SWB) is introduced in (Bonneel et al., 2015) by replacing Wasserstein distance with its sliced variant i.e., Sliced Wasseretein (SW) distance. Thank to the closed-form of Wasserstein distance in one-dimension, SWB has a low time complexity i.e., $\mathcal{O}(n \log n)$ which enables fast computation. Combining with the fact that Sliced Wasserstein is equivalent to Wasserstein distance in bounded domains (Bonnotte, 2013) and Sliced Wasserstein does not suffer from the curse of dimensionality (Nguyen et al., 2021; Nadjahi et al., 2020; Manole et al., 2022; Nietert et al., 2022), SWB becomes a scalable alternative choice of Wasserstein barycenter.

In some applications, we might want to find a barycenter that minimizes the distances to marginals while having equal distances to marginals at the same time e.g., constructing shape template for a group of shapes (Bongratz et al., 2022; Sun et al., 2023) that can be further used in downstream tasks, exact balance style mixing between images (Bonneel et al., 2015), fair generative modeling (Choi et al., 2020), and so on. We refer to such a barycenter as a marginal fairness barycenter. Both the Wasserstein barycenter and SWB are defined based on a given set of marginal weights (marginal coefficients), and these weights represent the importance levels of marginals toward the barycenter. Nevertheless, a uniform (weights) barycenter does not necessarily lead to the desired marginal fairness barycenter as shown in Figure 1. Moreover, obtaining the marginal fairness barycenter is challenging since such a barycenter might not exist and might not be identifiable given non-global-optimal optimization (Karcher mean problem). To the best of our knowledge, there is no prior work that investigates finding a marginal fairness barycenter.

In this work, we make the first attempt to tackle the marginal fairness barycenter problem i.e., we focus on finding Marginal Fairness Sliced Wasserstein Barycenter (MFSWB) to utilize the scalability of SW distance.

**Contribution:** In summary, our main contributions are four-fold:

1. We define the Marginal Fairness Sliced Wasserstein Barycenter (MFSWB) problem, which is a constrained barycenter problem where the constraint aims to limit the average pair-wise absolute difference between distances from the barycenter to the marginals. We derive the dual form of MFSWB, discuss its computation, and address its computational challenges.

2. To address this issue, we propose surrogate definitions of MFSWB that are hyperparameter-free and computationally tractable. Motivated by Fair PCA (Samadi et al., 2018), we propose the first surrogate MFSWB, which minimizes the largest SW distance from the barycenter to the marginals. To solve the problem of biased gradient estimation of the first surrogate MFSWB, we propose the second surrogate MFSWB, which is the expectation of the largest one-dimensional Wasserstein distance from the projected barycenter to the projected marginals. We show that the second surrogate is an upper bound of the first surrogate and can yield an unbiased gradient estimator. We further extend the second surrogate to the third surrogate by applying slicing distribution selection and show that the third surrogate is an upper bound of the previous two.

3. We discuss the connection between the proposed surrogate MFSWB problems and the Sliced Multi-marginal Wasserstein (SMW) distance with the maximal ground metric. In particular, solving

the proposed MFSWB problems is equivalent to minimizing a lower bound of the SMW. By showing that the SMW with the maximal ground metric is a generalized metric, we demonstrate that it is safe to use the proposed surrogate MFSWB problems.

4. We conduct simulations with Gaussian data and experiments on various applications, including 3D point-cloud averaging, color harmonization, and sliced Wasserstein autoencoder with class-fair representation, to demonstrate the favorable performance of the proposed surrogate definitions.

**Organization.** We first discuss some preliminaries on SW distance, SWB, its computation, and Sliced Multi-marginal Wasserstein distance in Section 2. We then introduce the formal definition and surrogate definitions of marginal fairness SWB in Section 3. Next, we conduct experiments to demonstrate the favorable performance and fairness of the proposed definitions in Section 4. We conclude the paper and provide some future directions in Section 5. Finally, we defer the proofs of key results, the discussion on related works, and additional materials to the Appendices.

## 2 PRELIMINARIES

**Sliced Wasserstein distance.** The definition of sliced Wasserstein (SW) distance (Bonneel et al., 2015) between two probability measures $\mu_1 \in \mathcal{P}_p(\mathbb{R}^d)$ and $\mu_2 \in \mathcal{P}_p(\mathbb{R}^d)$ is:

$$\text{SW}_p^p(\mu_1, \mu_2) = \mathbb{E}_{\theta \sim \mathcal{U}(\mathbb{S}^{d-1})}[\text{W}_p^p(\theta \sharp \mu_1, \theta \sharp \mu_2)], \tag{1}$$

where the Wasserstein distance has a closed form in one-dimension which is $\text{W}_p^p(\theta \sharp \mu_1, \theta \sharp \mu_2) = \int_0^1 |F_{\theta \sharp \mu_1}^{-1}(z) - F_{\theta \sharp \mu_2}^{-1}(z)|^p dz$ where $\theta \sharp \mu$ and $\theta \sharp \nu$ denotes the pushforward measures of $\mu$ and $\nu$ through the function $f(x) = \theta^\top x$, $F_{\theta \sharp \mu_1}$ and $F_{\theta \sharp \mu_2}$ are the cumulative distribution function (CDF) of $\theta \sharp \mu_1$ and $\theta \sharp \mu_2$ respectively.

**Sliced Wasserstein Barycenter.** The definition of the sliced Wasserstein barycenter (SWB) problem (Bonneel et al., 2015) of $K \geq 2$ marginals $\mu_1, \ldots, \mu_K \in \mathcal{P}_p(\mathbb{R}^d)$ with marginal weights $\omega_1, \ldots, \omega_K > 0$ ($\sum_{i=k}^K \omega_k = 1$) is defined as:

$$\min_\mu \mathcal{F}(\mu; \mu_{1:K}, \omega_{1:K}); \quad \mathcal{F}(\mu; \mu_{1:K}, \omega_{1:K}) = \sum_{k=1}^K \omega_k \text{SW}_p^p(\mu, \mu_k). \tag{2}$$

When $\omega_1 = \ldots = \omega_K = 1/K$, we obtain an uniform SWB problem.

**Computation of parametric SWB.** Let $\mu_\phi$ be parameterized by $\phi \in \Phi$, SWB can be solved by gradient-based optimization. In that case, the interested quantity is the gradient $\nabla_\phi \mathcal{F}(\mu_\phi; \mu_{1:K}, \omega_{1:K}) = \sum_{k=1}^K \omega_k \nabla_\phi \text{SW}_p^p(\mu_\phi, \mu_k)$. However, the gradient

$$\nabla_\phi \text{SW}_p^p(\mu_\phi, \mu_k) = \nabla_\phi \mathbb{E}_{\theta \sim \mathcal{U}(\mathbb{S}^{d-1})}[\text{W}_p^p(\theta \sharp \mu_\phi, \theta \sharp \mu_k)] = \mathbb{E}_{\theta \sim \mathcal{U}(\mathbb{S}^{d-1})}[\nabla_\phi \text{W}_p^p(\theta \sharp \mu_\phi, \theta \sharp \mu_k)]$$

for any $k = 1, \ldots, K$ is intractable due to the intractability of SW with the expectation with respect to the uniform distribution over the unit-hypersphere. Therefore, Monte Carlo estimation is used. In particular, projecting directions $\theta_1, \ldots, \theta_L$ are sampled i.i.d from $\mathcal{U}(\mathbb{S}^{d-1})$, and the stochastic gradient estimator is formed:

$$\nabla_\phi \text{SW}_p^p(\mu_\phi, \mu_k) \approx \frac{1}{L} \sum_{l=1}^L \nabla_\phi \text{W}_p^p(\theta_l \sharp \mu_\phi, \theta_l \sharp \mu_k). \tag{3}$$

With the stochastic gradient, the SWB can be solved by using a stochastic gradient descent algorithm. We refer the reader to Algorithm 1 in Appendix B for more detail. Specifically, we now discuss the discrete SWB i.e., marginals and the barycenter are discrete measures.

*Free supports barycenter.* In this setting, we have $\mu_\phi = \frac{1}{n} \sum_{i=1}^n \delta_{x_i}$, $\mu_k = \frac{1}{n} \sum_{i=1}^n \delta_{y_i}$, and $\phi = (x_1, \ldots, x_n)$, we can compute the (sub-)gradient with the time complexity $\mathcal{O}(n \log n)$:

$$\nabla_{x_i} \text{W}_p^p(\theta \sharp \mu_\phi, \theta \sharp \mu_k) = p|\theta^\top x_i - \theta^\top y_{\sigma(i)}|^{p-1} \text{sign}(\theta^\top x_i - \theta^\top y_{\sigma(i)})\theta, \tag{4}$$

where $\sigma = \sigma_1 \circ \sigma_2^{-1}$ with $\sigma_1$ and $\sigma_2$ are any sorted permutation of $\{x_1, \ldots, x_n\}$ and $\{y_1, \ldots, y_n\}$. Here, $[n]$ denotes the set $\{1, 2, \ldots, n\}$, $\sigma_1 : [n] \to [n]$ is the permuation function such that $x_{\sigma_1(1)} \leq$

$x_{\sigma_1(2)} \leq \ldots \leq x_{\sigma_1(n)}$ or $x_{\sigma_1(1)} \geq x_{\sigma_1(2)} \geq \ldots \geq x_{\sigma_1(n)}$. Similarly, $\sigma_2 : [n] \to [n]]$ is the permuation function such that $y_{\sigma_2(1)} \leq y_{\sigma_2(2)} \leq \ldots \leq y_{\sigma_2(n)}$ or $y_{\sigma_2(1)} \geq y_{\sigma_2(2)} \geq \ldots \geq y_{\sigma_2(n)}$, and $\sigma_2^{-1}$ is the argsort operator. The transport map is contructed as $\sigma = \sigma_1 \circ \sigma_2^{-1}$.

*Fixed supports barycenter.* In this setting, we have $\mu_\phi = \sum_{i=1}^n \phi_i \delta_{x_i}$, $\mu_k = \sum_{i=1}^n \beta_i \delta_{x_i}$, $\sum_{i=1}^n \phi_i = \sum_{i=1}^n \beta_i$ and $\phi = (\phi_1, \ldots, \phi_n)$. We can compute the gradient as follows:

$$\nabla_\phi W_p^p(\theta \sharp \mu_\phi, \theta \sharp \mu_k) = \boldsymbol{f}^\star, \tag{5}$$

where $\boldsymbol{f}^\star$ is the first optimal Kantorovich dual potential of $W_p^p(\theta \sharp \mu_\phi, \theta \sharp \mu_k)$ which can be obtained with the time complexity of $\mathcal{O}(n \log n)$. We refer the reader to Proposition 1 in (Cuturi & Doucet, 2014) for the detail and Algorithm 1 in (Séjourné et al., 2022) for the computational algorithm.

When the supports or weights of the barycenter are the output of a parametric function, we can use the chain rule to estimate the gradient of the parameters of the function. For the continuous case, we can approximate the barycenter and marginals by their empirical versions, and then perform the estimation in the discrete case. Since the sample complexity of SW is $\mathcal{O}(n^{-1/2})$ (Nadjahi et al., 2019; Nguyen et al., 2021; Manole et al., 2022; Nietert et al., 2022), the approximation error will reduce fast with the number of support $n$ increases. Another option is to use continuous Wasserstein solvers (Fan et al., 2021; Korotin et al., 2022; Claici et al., 2018), however, this option is not as simple as the first one.

**Sliced Multi-marginal Wasserstein Distance.** Given $K \geq 1$ marginals $\mu_1, \ldots, \mu_K \in \mathcal{P}_p(\mathbb{R}^d)$, Sliced Multi-marginal Wasserstein Distance (Cohen et al., 2021) (SMW) is defined as:

$$SMW_p^p(\mu_{1:K}; c) = \mathbb{E}\left[\inf_{\pi \in \Pi(\mu_1, \ldots, \mu_K)} \int c(\theta^\top x_1, \ldots, \theta^\top x_K)^p d\pi(x_1, \ldots, x_K)\right], \tag{6}$$

where the expectation is under $\theta \sim \mathcal{U}(\mathbb{S}^{d-1})$. When using the barycentric cost i.e.,

$$c(\theta^\top x_1, \ldots, \theta^\top x_K)^p = \sum_{k=1}^K \beta_k \left| \theta^\top x_k - \sum_{k'=1}^K \beta_{k'} \theta^\top x_{k'} \right|^p,$$

for $\beta_k > 0 \quad \forall k$ and $\sum_k \beta_k = 1$. Minimizing $SMW_p^p(\mu_{1:K}, \mu; c)$ with respect to $\mu$ is equivalent to a barycenter problem. We refer the reader to Proposition 7 in (Cohen et al., 2021) for more detail.

# 3 MARGINAL FAIRNESS SLICED WASSERSTEIN BARYCENTER

We first formally define the marginal fairness Sliced Wasserstein barycenter in Section 3.1. We then propose surrogate problems in Section 3.2. Finally, we discuss the connection of the proposed surrogate problems to sliced multi-marginal Wasserstein in Section 3.3.

## 3.1 FORMAL DEFINITION

Now, we define the Marginal Fairness Sliced Wasserstein Barycenter (MFSWB) problem by adding marginal fairness constraints to the SWB problem.

**Definition 1.** *Given $K \geq 2$ marginals $\mu_1, \ldots, \mu_K \in \mathcal{P}_p(\mathbb{R}^d)$, admissible $\epsilon \geq 0$ for $i = 1, \ldots, K$ and $j = i + 1, \ldots, K$, the Marginal Fairness Sliced Wasserstein barycenter (MFSWB) is defined as:*

$$\min_\mu \frac{1}{K} \sum_{k=1}^K SW_p^p(\mu, \mu_k) \quad s.t. \quad \frac{2}{(K-1)K} \sum_{i=1}^{K-1} \sum_{j=i+1}^K |SW_p^p(\mu, \mu_i) - SW_p^p(\mu, \mu_j)| \leq \epsilon. \tag{7}$$

**Remark 1.** *We want $\epsilon$ in Definition 1 to be close to 0 i.e., $\mu_1, \ldots, \mu_K$ are on the $SW_p$-sphere with the center $\mu$. However, for a too-small value of $\epsilon$, there might not exist a solution $\mu$.*

*Duality objective.* For admissible $\epsilon > 0$, there exist a Lagrange multiplier $\lambda$ such that we have the dual form

$$\mathcal{L}(\mu, \lambda) = \frac{1}{K} \sum_{k=1}^K SW_p^p(\mu, \mu_k) + \frac{2\lambda}{(K-1)K} \sum_{i=1}^{K-1} \sum_{j=i+1}^K |SW_p^p(\mu, \mu_i) - SW_p^p(\mu, \mu_j)| - \lambda \epsilon. \tag{8}$$

*Computational challenges.* Firstly, MFSWB in Definition 1 requires an admissible $\epsilon > 0$ to guarantee the existence of the barycenter $\mu$. In practice, it is unknown if a value of $\epsilon$ satisfies such a property. Secondly, given an $\epsilon$, it is not trivial to obtain the optimal Lagrange multiplier $\lambda^\star$ in Equation equation 8 to minimize the duality gap, which can be non-zero (weak duality). Thirdly, directly using the dual objective in Equation equation 8 requires hyperparameter tuning for $\lambda$ and might not provide a good landscape for optimization. Moreover, we cannot obtain an unbiased gradient estimate of $\phi$ in the case of the parametric barycenter $\mu_\phi$. In greater detail, the Monte Carlo estimation of the absolute distance between two SW distances is biased. Finally, Equation equation 8 has a quadratic time complexity and space complexity in terms of the number of marginals, i.e., $\mathcal{O}(K^2)$.

## 3.2 SURROGATE DEFINITIONS

Since it is not convenient to use the formal MFSWB in applications, we propose three surrogate definitions of MFSWB that are free of hyperparameters and computationally friendly.

**First Surrogate Definition.** Motivated by Fair PCA (Samadi et al., 2018), we propose a practical surrogate MFSWB problem that is hyperparameter-free.

**Definition 2.** *Given $K \geq 2$ marginals $\mu_1, \ldots, \mu_K \in \mathcal{P}_p(\mathbb{R}^d)$, the surrogate Marginal Fairness Sliced Wasserstein Barycenter (s-MFSWB) problem is defined as:*

$$\min_\mu \mathcal{SF}(\mu; \mu_{1:K}); \quad \mathcal{SF}(\mu; \mu_{1:K}) = \max_{k \in \{1, \ldots, K\}} SW_p^p(\mu, \mu_k). \tag{9}$$

The s-MFSWB problem tries to minimize the maximal distance from the barycenter to the marginals. Therefore, it can minimize indirectly the overall distances between the barycenter to the marginals and implicitly make the distances to marginals approximately the same.

*Gradient estimator.* Let $\mu_\phi$ be paramterized by $\phi \in \Phi$, and $\mathcal{F}(\phi, k) = SW_p^p(\mu_\phi, \mu_k)$, we would like to compute $\nabla_\phi \max_{k \in \{1, \ldots, K\}} \mathcal{F}(\phi, k)$. By Danskin's envelope theorem (Danskin, 2012), we have:

$$\nabla_\phi \max_{k \in \{1, \ldots, K\}} \mathcal{F}(\phi, k) = \nabla_\phi \mathcal{F}(\phi, k^\star) = \nabla_\phi SW_p^p(\mu_\phi, \mu_{k^\star}),$$

for $k^\star = \arg\max_{k \in \{1, \ldots, K\}} \mathcal{F}(\phi, k)$. Nevertheless, $k^\star$ is intractable due to the intractablity of $SW_p^p(\mu_\phi, \mu_k)$ for $k = 1, \ldots, K$. Hence, we can form the estimation

$$\hat{k}^\star = \arg\max_{k \in \{1, \ldots, K\}} \widehat{SW}_p^p(\mu_\phi, \mu_k; L)$$

where $\widehat{SW}_p^p(\mu_\phi, \mu_k; L) = \frac{1}{L} \sum_{l=1}^L W_p^p(\theta_l \sharp \mu_\phi, \theta_l \sharp \mu_k)$ with $\theta_1, \ldots, \theta_L \overset{i.i.d}{\sim} \mathcal{U}(\mathbb{S}^{d-1})$. Then, we can estimate $\nabla_\phi SW_p^p(\mu_\phi, \mu_{\hat{k}^\star})$ as in Equation 3. We refer the reader to Algorithm 2 in Appendix B for the gradient estimation and optimization procedure. The downside of this estimator is that it is biased.

**Second Surrogate Definition.** To address the biased gradient issue of the first surrogate problem, we propose the second surrogate MFSWB problem.

**Definition 3.** *Given $K \geq 2$ marginals $\mu_1, \ldots, \mu_K \in \mathcal{P}_p(\mathbb{R}^d)$, the unbiased surrogate Marginal Fairness Sliced Wasserstein Barycenter (us-MFSWB) problem is defined as:*

$$\min_\mu \mathcal{USF}(\mu; \mu_{1:K}); \quad \mathcal{USF}(\mu; \mu_{1:K}) = \mathbb{E}_{\theta \sim \mathcal{U}(\mathbb{S}^{d-1})} \left[ \max_{k \in \{1, \ldots, K\}} W_p^p(\theta \sharp \mu, \theta \sharp \mu_k) \right]. \tag{10}$$

In contrast to s-MFSWB which minimizes the maximal SW distance among marginals, us-MFSWB minimizes the expected value of the maximal one-dimensional Wasserstein distance among marginals. By considering fairness on one-dimensional projections, us-MFSWB can yield an unbiased gradient estimate which is the reason why it is named as unbiased s-MFSWB.

*Gradient estimator.* Let $\mu_\phi$ be paramterized by $\phi \in \Phi$, and $\mathcal{F}(\theta, \phi, k) = W_p^p(\theta \sharp \mu_\phi, \theta \sharp \mu_k)$, we would like to compute $\nabla_\phi \mathbb{E}_{\theta \sim \mathbb{S}^{d-1}}[\max_{k \in \{1, \ldots, K\}} \mathcal{F}(\theta, \phi, k)]$ which is equivalent to $\mathbb{E}_{\theta \sim \mathbb{S}^{d-1}}[\nabla_\phi \max_{k \in \{1, \ldots, K\}} \mathcal{F}(\theta, \phi, k)]$ due to the Leibniz's rule. By Danskin's envelope theorem, we have:

$$\nabla_\phi \max_{k \in \{1, \ldots, K\}} \mathcal{F}(\theta, \phi, k) = \nabla_\phi \mathcal{F}(\theta, \phi, k^\star) = \nabla_\phi W_p^p(\theta \sharp \mu_\phi, \theta \sharp \mu_{k^\star}),$$

for $k_\theta^\star = \arg\max_{k \in \{1,\dots,K\}} \mathcal{F}(\theta, \phi, k)$ where we can estimate $\nabla_\phi W_p^p(\theta \sharp \mu_\phi, \theta \sharp \mu_{k_\theta^\star})$ can be computed as in Equation 4- 5. Overall, with $\theta_1, \dots, \theta_L \overset{i.i.d}{\sim} \mathcal{U}(\mathbb{S}^{d-1})$, we can form the final estimation $\frac{1}{L} \sum_{l=1}^L \nabla_\phi W_p^p(\theta_l \sharp \mu_\phi, \theta_l \sharp \mu_{k_{\theta_l}^\star})$ which is an unbiased estimate. We refer the reader to Algorithm 3 in Appendix B for the gradient estimation and optimization procedure.

**Proposition 1.** *Given $K \geq 2$ marginals $\mu_{1:K} \in \mathcal{P}_p(\mathbb{R}^d)$, we have $\mathcal{SF}(\mu; \mu_{1:K}) \leq \mathcal{USF}(\mu; \mu_{1:K})$.*

Proof of Proposition 1 is given in Appendix A.1. From the proposition, we see that minimizing the objective of us-MFSWB also reduces the objective of s-MFSWB implicitly.

**Proposition 2.** *Given $K \geq 2$ marginals $\mu_1, \dots, \mu_K \in \mathcal{P}_p(\mathbb{R}^d)$, $\theta_1, \dots, \theta_L \overset{i.i.d}{\sim} \mathcal{U}(\mathbb{S}^{d-1})$, we have:*

$$\mathbb{E}\left|\nabla_\phi \frac{1}{L} \sum_{l=1}^L W_p^p(\theta_l \sharp \mu_\phi, \theta_l \sharp \mu_{k_\theta^\star}) - \nabla_\phi \mathcal{USF}(\mu_\phi; \mu_{1:K})\right| \leq \frac{1}{\sqrt{L}} Var\left[\nabla_\phi W_p^p(\theta \sharp \mu_\phi, \theta \sharp \mu_{k_\theta^\star})\right]^{\frac{1}{2}}, \quad (11)$$

*where $k_\theta^\star = \arg\max_{k \in \{1,\dots,K\}} W_p^p(\theta \sharp \mu_\phi, \theta \sharp \mu_k)$; and the expectation and variance are under the random projecting direction $\theta \sim \mathcal{U}(\mathbb{S}^{d-1})$*

Proof of Proposition 2 is given in Appendix A.2. From the proposition, we know that the approximation error of the gradient estimator of us-MFSWB reduces at the order of $\mathcal{O}(L^{-1/2})$. Therefore, increasing $L$ leads to a better gradient approximation. The approximation could be further improved via Quasi-Monte Carlo methods (Nguyen et al., 2024a).

**Third Surrogate Definition.** The us-MFSWB in Definition 3 utilizes the uniform distribution as the slicing distribution, which is empirically shown to be non-optimal in statistical estimation (Nguyen et al., 2021). Following the slicing distribution selection approach in (Nguyen & Ho, 2023), we propose the third surrogate with a new slicing distribution that focuses on unfair projecting directions.

*Marginal Fairness energy-based Slicing distribution.* Since we want to encourage marginal fairness, it is natural to construct the slicing distribution based on fairness energy.

**Definition 4.** *Given $K \geq 2$ marginals $\mu_1, \dots, \mu_K \in \mathcal{P}_p(\mathbb{R}^d)$, the Marginal Fairness energy-based Slicing distribution $\sigma(\theta; \mu, \mu_{1:K}) \in \mathcal{P}(\mathbb{S}^{d-1})$ is defined with the density function as follow:*

$$f_\sigma(\theta; \mu, \mu_{1:K}) \propto \exp\left(\max_{k \in \{1,\dots,K\}} W_p^p(\theta \sharp \mu, \theta \sharp \mu_k)\right), \quad (12)$$

We see that the marginal fairness energy-based slicing distribution in Definition 4 put more mass to a projecting direction $\theta$ that has the larger maximal one-dimensional Wasserstein distance to marginals. Therefore, it will penalize more marginally unfair projecting directions.

*Energy-based surrogate MFSWB.* From the new proposed slicing distribution, we can define a new surrogate MFSWB problem, named Energy-based surrogate MFSWB.

**Definition 5.** *Given $K \geq 2$ marginals $\mu_1, \dots, \mu_K \in \mathcal{P}_p(\mathbb{R}^d)$, the energy-based surrogate Marginal Fairness Sliced Wasserstein Barycenter (es-MFSWB) problem is defined as:*

$$\min_\mu \mathcal{ESF}(\mu; \mu_{1:K}); \quad \mathcal{ESF}(\mu; \mu_{1:K}) = \mathbb{E}_{\theta \sim \sigma(\theta; \mu, \mu_{1:K})}\left[\max_{k \in \{1,\dots,K\}} W_p^p(\theta \sharp \mu, \theta \sharp \mu_k)\right]. \quad (13)$$

Similar to the us-MFSWB, es-MFSWB also employs the implicit one-dimensional marginal fairness. Nevertheless, es-MFSWB utilizes the marginal fairness energy-based slicing distribution to reweight the importance of each projecting direction instead of treating them equally.

**Proposition 3.** *Given $K \geq 2$ marginals $\mu_{1:K} \in \mathcal{P}_p(\mathbb{R}^d)$, we have $\mathcal{USF}(\mu; \mu_{1:K}) \leq \mathcal{ESF}(\mu; \mu_{1:K})$.*

Proof of Proposition 3 is given in Appendix A.3. According to the proposition, we see that minimizing the objective of es-MFSWB implicitly reduces the objective of us-MFSWB thereby decreasing the objective of s-MFSWB as well (Proposition 1)."

*Gradient estimator.* Let $\mu_\phi$ be parameterized by $\phi \in \Phi$, we want to estimate $\nabla_\phi \mathcal{ESF}(\mu_\phi; \mu_{1:K})$. Since the slicing distribution is unnormalized, we use importance sampling to form an estimation.

With $\theta_1, \ldots, \theta_L \overset{i.i.d}{\sim} \mathcal{U}(\mathbb{S}^{d-1})$, we can form the importance sampling stochastic gradient estimation:

$$\hat{\nabla}_\phi \mathcal{ESF}(\mu_\phi; \mu_{1:K}, L) = \frac{1}{L} \sum_{l=1}^{L} \left[ \nabla_\phi \left( W_p^p(\theta_l \sharp \mu, \theta_l \sharp \mu_{k_{\theta_l}^\star}) \frac{\exp\left(W_p^p(\theta_l \sharp \mu, \theta_l \sharp \mu_{k_{\theta_l}^\star})\right)}{\frac{1}{L} \sum_{i=1}^{L} \left[\exp\left(W_p^p(\theta_i \sharp \mu, \theta_i \sharp \mu_{k_{\theta_i}^\star})\right)\right]} \right) \right],$$

which can be further derived by using the chain rule and previously discussed techniques. It is worth noting that the above estimation is only asymptotically unbiased. We refer the reader to Algorithm 4 in Appendix B for the gradient estimation and optimization procedure.

**Computational complexities of proposed surrogates.** For the number of marginals $K$, the three proposed surrogates have a linear time complexity and space complexity i.e., $\mathcal{O}(K)$ which is the same as the conventional SWB and is better than $\mathcal{O}(K^2)$ of the formal MFSWB. For the number of projections $L$, the number of supports $n$, and the number of dimensions $d$, the proposed surrogates have the time complexity of $\mathcal{O}(Ln(\log n + d))$ and the space complexity of $\mathcal{O}(L(n + d))$ which are similar to the formal MFSWB and SWB.

### 3.3 SLICED MULTI-MARGINAL WASSERSTEIN DISTANCE WITH MAXIMAL GROUND METRIC

To shed some light on the proposed substrates, we connect them to a special variant of Sliced multi-marginal Wasserstein (SMW) (see Equation 6) i.e., SMW with the maximal ground metric

$$c(\theta^\top x_1, \ldots, \theta^\top x_K) = \max_{i \in \{1, \ldots, K\}, j \in \{1, \ldots, K\}} |\theta^\top x_i - \theta^\top x_j|.$$

We first show that SMW with the maximal ground metric is a generalized metric on the space of probability measures.

**Proposition 4.** *Sliced multi-marginal Wasserstein distance with the maximal ground metric is a generalized metric i.e., it satisfies non-negativity, marginal exchangeability, generalized triangle inequality, and identity of indiscernibles.*

Proof of Proposition 4 is given in Appendix A.4. It is worth noting that SMW with the maximal ground metric has never been defined before. Since our work focuses on the MFSWB problem, we will leave the careful investigation of this variant of SMW to future work.

**Proposition 5.** *Given $K \geq 2$ marginals $\mu_1, \ldots, \mu_K \in \mathcal{P}_p(\mathbb{R}^d)$, the maximal ground metric $c(\theta^\top x_1, \ldots, \theta^\top x_K) = \max_{i \in \{1, \ldots, K\}, j \in \{1, \ldots, K\}} |\theta^\top x_i - \theta^\top x_j|$, we have:*

$$\min_{\mu_1} \mathcal{USF}(\mu_1; \mu_{2:K}) \leq \min_{\mu_1} SMW_p^p(\mu_1, \mu_2, \ldots, \mu_K; c). \tag{14}$$

Proof of Proposition 5 is given in Appendix A.5 and the inequality holds when changing $\mu_1$ to any $\mu_i$ with $i = 2, \ldots, K$. Combining Proposition 1, we have the corollary of $\min_{\mu_1} \mathcal{SF}(\mu_1; \mu_{2:K}) \leq \min_{\mu_1} SMW_p^p(\mu_1, \mu_2, \ldots, \mu_K; c)$. From the proposition, we see that minimizing the us-MFSWB is equivalent to minimizing a lower bound of SMW with the maximal ground metric. Therefore, this proposition implies the us-MFSWB could try to minimize the multi-marginal distance. Moreover, this proposition can help to understand the proposed surrogates through the gradient flow of SMW. We can further extend the proposition to show the minimizing es-MFSWB objective is the same as minimizing a lower bound of energy-based SMW with the maximal ground metric, a new special variant of SMW. We refer the reader to Propositon 6 in Appendix B for more detail.

## 4 EXPERIMENTS

In this section, we compare the barycenter found by our proposed surrogate problems i.e., s-MFSWB, us-MFSWB, and es-MFSWB with the barycenter found by USWB and the formal MFSWB. For evaluation, we use two metrics i.e., the F-metric (F) and the W-metric (W) which are defined as follows:

$$F = \frac{2}{K(K-1)} \sum_{i=1}^{K-1} \sum_{j=i+1}^{K} |W_p^p(\mu, \mu_i) - W_p^p(\mu, \mu_j)|, \quad W = \frac{1}{K} \sum_{i=1}^{K} W_p^p(\mu, \mu_i),$$

where $\mu$ is the barycenter, $\mu_1, \ldots, \mu_K$ are the given marginals, and $W_p^p$ is the Wasserstein distance (Flamary et al., 2021) of the order $p$. Here, the F-metric represents the marginal fairness degree of the barycenter and the W-metric represents the centrality of the barycenter. For all following experiments, we use $p = 2$ for the Wasserstein distance and barycenter problems.

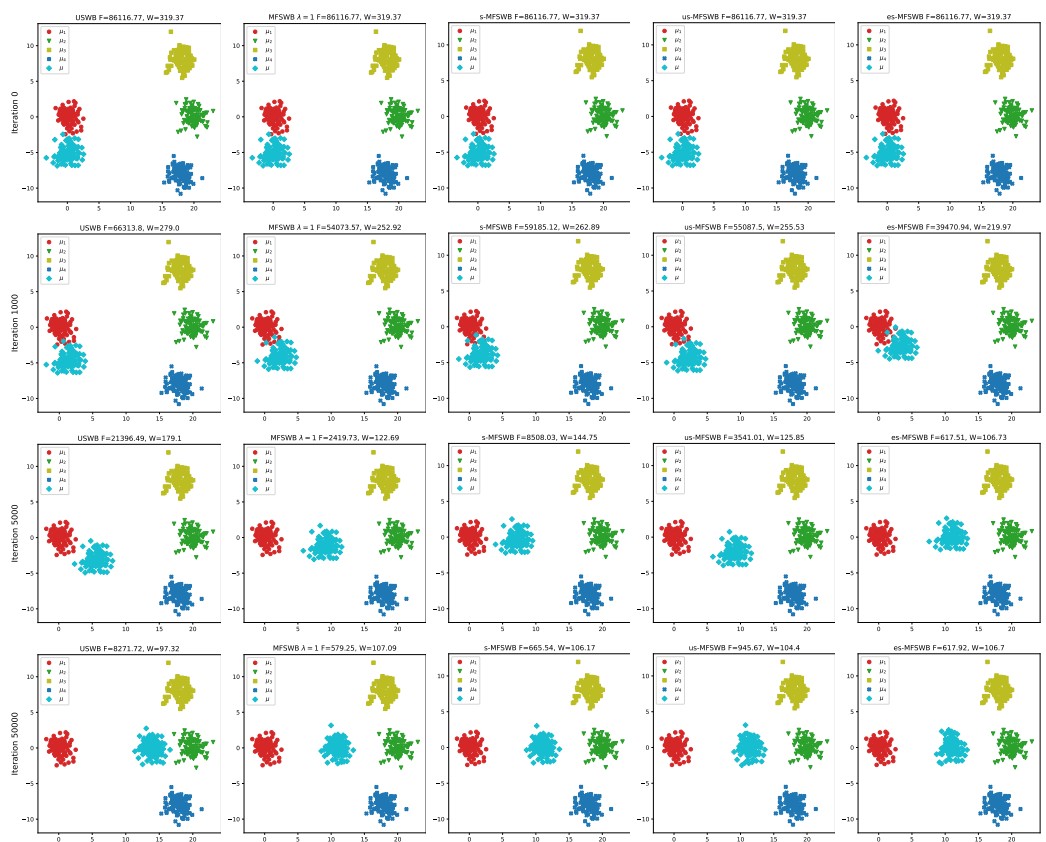

Figure 2: Barycenters from USWB, MFSWB with $\lambda = 1$, s-MFSWB, us-MFSWB, and es-MFSWB along gradient iterations with the corresponding F-metric and W-metric.

### 4.1 BARYCENTER OF GAUSSIANS

We first start with a simple simulation with 4 marginals which are empirical distributions with 100 i.i.d samples from 4 Gaussian distributions i.e., $\mathcal{N}((0,0), I)$, $\mathcal{N}((20,0), I)$, $\mathcal{N}((18,8), I)$, and $\mathcal{N}((18,-8), I)$. We then find the barycenter which is represented as an empirical distribution with 100 supports initialized by sampling i.i.d from $\mathcal{N}((0,-5), I)$. We use stochastic gradient descent with 50000 iterations of learning rate 0.01, the number of projections 100. We show the visualization of the found barycenters with the corresponding F-metric and W-metric by using USWB, s-MFSWB, us-MFSWB, and es-MFSWB at iterations 0, 1000, 5000, and 50000 in Figure 2. We observe that the USWB does not lead to a marginal fairness barycenter. The three proposed surrogate problems help to find a better barycenter faster in both two metrics than USWB. At convergence i.e., iteration 50000, we see that USWB does not give a fair barycenter while the three proposed surrogates lead to a more fair barycenter. Among the proposed surrogates, es-MFSWB gives the most marginal fairness barycenter with a competitive centerness. The formal MFSWB (dual form with $\lambda = 1$) leads to the most fair barycenter. However, the performance of the formal MFSWB is quite sensitive to $\lambda$. We also observe the same phenomenon for different choices of learning rate in Figure 5 in Appendix D. We show the visualization for $\lambda = 0.1$ and $\lambda = 10$ in Figure 6 in Appendix D.

### 4.2 3D POINT-CLOUD AVERAGING

We aim to find the mean shape of point-cloud shapes by casting a point cloud $X = \{x_1, \ldots, x_n\}$ into an empirical probability measures $P_X = \frac{1}{n} \sum_{i=1}^{n} \delta_{x_i}$. We select two point-cloud shapes which consist of 2048 points in ShapeNet Core-55 dataset (Chang et al., 2015). We initialize the barycenter with a spherical point-cloud. We use stochastic gradient descent with 10000 iterations of learning rate 0.01, the number of projections 10. We report the found barycenters for two car shapes in Figure 3 at the final iteration and the corresponding F-metric and W-metric at iterations 0, 1000, 5000, and 10000

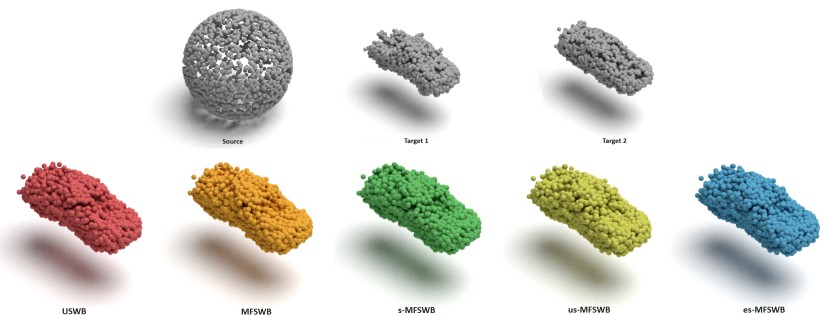

Figure 3: Averaging point-clouds with USWB, MFSWB ($\lambda = 1$), s-MFSWB, us-MFSWB, and es-MFSWB.

Table 1: F-metric and W-metric along iterations in point-cloud averaging application.

| Method | Iteration 0 | | Iteration 1000 | | Iteration 5000 | | Iteration 10000 | |
|---|---|---|---|---|---|---|---|---|
| | F ($\downarrow$) | W ($\downarrow$) | F ($\downarrow$) | W ($\downarrow$) | F ($\downarrow$) | W ($\downarrow$) | F ($\downarrow$) | W ($\downarrow$) |
| USWB | $252.24 \pm 0.0$ | $3746.05 \pm 0.0$ | $4.89 \pm 0.28$ | $85.72 \pm 0.18$ | $3.79 \pm 0.32$ | $45.37 \pm 0.18$ | $1.55 \pm 0.48$ | $39.81 \pm 0.18$ |
| MFSWB $\lambda = 0.1$ | $252.24 \pm 0.0$ | $3746.05 \pm 0.0$ | $4.76 \pm 0.27$ | $84.86 \pm 0.17$ | $3.78 \pm 0.2$ | $45.2 \pm 0.11$ | $1.32 \pm 0.22$ | $39.73 \pm 0.16$ |
| MFSWB $\lambda = 1$ | $252.24 \pm 0.0$ | $3746.05 \pm 0.0$ | $0.49 \pm 0.2$ | $79.08 \pm 0.15$ | $3.64 \pm 0.26$ | $44.71 \pm 0.19$ | $1.03 \pm 0.06$ | $39.45 \pm 0.18$ |
| MFSWB $\lambda = 10$ | $252.24 \pm 0.0$ | $3746.05 \pm 0.0$ | $4.03 \pm 2.43$ | $\mathbf{71.24 \pm 0.9}$ | $7.32 \pm 2.5$ | $45.21 \pm 0.2$ | $4.13 \pm 2.48$ | $42.56 \pm 0.36$ |
| s-MFSWB | $252.24 \pm 0.0$ | $3746.05 \pm 0.0$ | $2.52 \pm 0.77$ | $81.84 \pm 0.14$ | $4.01 \pm 0.38$ | $44.9 \pm 0.13$ | $1.15 \pm 0.09$ | $39.58 \pm 0.17$ |
| us-MFSWB | $252.24 \pm 0.0$ | $3746.05 \pm 0.0$ | $0.3 \pm 0.18$ | $78.69 \pm 0.17$ | $3.74 \pm 0.26$ | $44.38 \pm 0.1$ | $0.87 \pm 0.18$ | $39.26 \pm 0.1$ |
| es-MFSWB | $252.24 \pm 0.0$ | $3746.05 \pm 0.0$ | $\mathbf{0.2 \pm 0.19}$ | $78.1 \pm 0.16$ | $\mathbf{3.5 \pm 0.29}$ | $\mathbf{44.37 \pm 0.08}$ | $\mathbf{0.84 \pm 0.22}$ | $\mathbf{39.18 \pm 0.08}$ |

in Table 1 from three independent runs. As in the Gaussian simulation, s-MFSWB, us-MFSWB, and es-MFSWB help to reduce the two metrics faster than the USWB. With the slicing distribution selection, es-MFSWB performs the best at every iteration, even better than the formal MFSWB with three choices of $\lambda$ i.e., $0.1, 1, 10$. We also observe a similar phenomenon for two plane shapes in Figure 7 and Table 3 in Appendix D. We refer the reader to Appendix D for a detailed discussion.

## 4.3 COLOR HARMONIZATION

We want to transform the color palette of a source image, denoted as $X = (x_1, \ldots, x_n)$ for $n$ is the number of pixels, to be an exact hybrid between two target images. Similar to the previous point-cloud averaging, we transform the color palette of an image into the empirical probability measure over colors (RGB) i.e., $P_X = \frac{1}{n} \sum_{i=1}^{n} \delta_{x_i}$. We then minimize barycenter losses i.e., USWB, MFSWB ($\lambda \in \{0.1, 1, 10\}$), s-MFSWB, us-MFSWB, and es-MFSWB by using stochastic gradient descent with the learning rate $0.0001$ and $20000$ iterations. We report both the transformed images and the corresponding F-metric and W-metric in Figure 4. We also report the full results in Figure 8- 10 in Appendix D. As in previous experiments, we see that the three proposed surrogates yield a better barycenter faster than USWB. The proposed es-MFSWB is the best variant among all surrogates since it has the lowest F-metric and W-metric at all iterations. We refer the reader to Figure 11-Figure 14 in Appendix D for additional flowers-images example, where a similar relative comparison happens. For the formal MFSWB, it is worse than es-MFSWB in one setting and better than es-MFSWB in one setting with the right choice of $\lambda$. Therefore, it is more convenient to use us-MFSWB in practice.

## 4.4 SLICED WASSERSTEIN AUTOENCODER WITH CLASS-FAIR REPRESENTATION

**Problem.** We consider training the sliced Wasserstein autoencoder (SWAE)(Kolouri et al., 2018) with a class-fairness regularization. In particular, we have the data distributions of $K \geq 1$ classes i.e., $\mu_k \in \mathcal{P}(\mathbb{R}^d)$ for $k = 1, \ldots, K$ and we would like to estimate an encoder network $f_\phi : \mathbb{R}^d \to \mathbb{R}^h$ ($\phi \in \Phi$) and a decoder network $g_\psi : \mathbb{R}^h \to \mathbb{R}^d$ ($\psi \in \Psi$ with $\mathbb{R}^h$ is a low-dimensional latent space). Given a prior distribution $\mu_0 \in \mathcal{P}(\mathbb{R}^h)$, $p \geq 1$, $\kappa_1 \in \mathbb{R}^+$, $\kappa_2 \in \mathbb{R}^+$, and a minibatch size $M \geq 1$, we perform the following optimization problem:

$$\min_{\phi, \psi} \mathbb{E} \left[ \frac{1}{KM} \sum_{k=1}^{K} \sum_{i=1}^{M} c(X_{ki}, g_\psi(f_\phi(X_{ki}))) + \kappa_1 SW_p^p(P_Z, P_{(f_\phi(X_k))_{k=1}^K}) + \kappa_2 \mathcal{B}(P_Z; P_{f_\phi(X_1)} : P_{f_\phi(X_K)}) \right],$$

where $(X_1, \ldots, X_K) \sim \mu_1^{\otimes M} \otimes \ldots \otimes \mu_K^{\otimes M}$, $Z \sim \mu_0^{\otimes M}$, $c$ is a reconstruction loss, $P_Z = \frac{1}{M} \sum_{i=1}^{M} \delta_{Z_i}$, $P_{(f_\phi(X_k))_{k=1}^K} = \frac{1}{KM} \sum_{k=1}^{K} \sum_{i=1}^{M} \delta_{f_\phi(X_{ki})}$, $P_{f_\phi(X_k)} = \frac{1}{M} \sum_{i=1}^{M} \delta_{f_\phi(X_{ki})}$ for $k = 1, \ldots, K$, and $\mathcal{B}$ denotes a barycenter loss i.e., USWB, MFSWB, s-MFSWB, us-MFSWB, and es-MFSWB. This

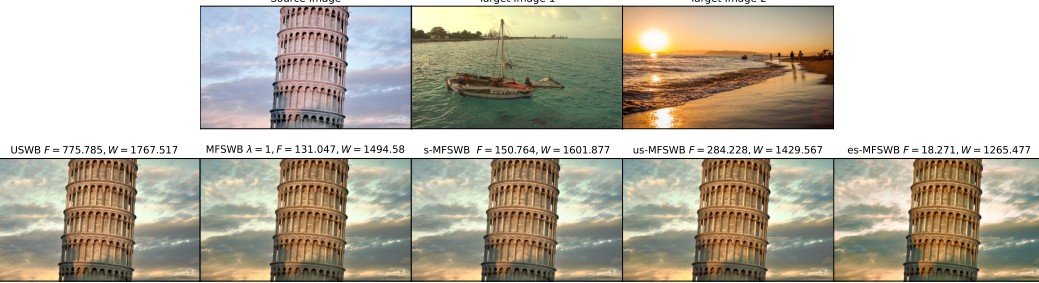

Figure 4: Harmonized images from USWB, MFSWB ($\lambda = 1$), s-MFSWB, us-MFSWB, and es-MFSWB.

Table 2: Results of grid search for learning rates in $\{0.0001, 0.0005, 0.001\}$ for training SWAE.

| Methods | RL ($\downarrow$) | $W_{2,\text{latent}}^2 \times 10^2$ ($\downarrow$) | $W_{2,\text{image}}^2 \times 10^2$ ($\downarrow$) | $F \times 10^2$ ($\downarrow$) | $W \times 10^2$ ($\downarrow$) | $F_{\text{images}}$ ($\downarrow$) |
|---|---|---|---|---|---|---|
| SWAE | 3.002 | 9.949 | **26.572** | 17.661 | 28.512 | 7.787 |
| USWB | 3.195 | 9.174 | 27.446 | 5.190 | 12.448 | 7.140 |
| MFSWB $\lambda = 0.1$ | **2.812** | 8.981 | 26.636 | 17.206 | 28.734 | 7.846 |
| MFSWB $\lambda = 1.0$ | 2.883 | 7.978 | 26.355 | 18.069 | 29.701 | 7.367 |
| MFSWB $\lambda = 10.0$ | 3.801 | 8.497 | 26.658 | 18.501 | 28.768 | 7.950 |
| s-MFSWB | 3.170 | **7.806** | 28.277 | 2.037 | 8.699 | 7.419 |
| us-MFSWB | 2.833 | 8.720 | 27.939 | 2.072 | **7.780** | **6.898** |
| es-MFSWB | 3.056 | 9.154 | 28.012 | **1.760** | 7.268 | 7.485 |

setting can be seen as an inverse barycenter problem i.e., the barycenter is fixed and the marginals are learnt under some constraints (e.g., the reconstruction loss and the aggregated distribution loss).

**Results.** We train the autoencoder on MNIST dataset (LeCun et al., 1998) ($d = 28 \times 28$) with $\kappa_1 = 8.0$, $\kappa_2 = 0.5$, 250 epochs, using a uniform distribution on a 2D ball ($h = 2$) as $\mu_0$ with differnt learning rates: $\{0.0001, 0.0005, 0.0008, 0.001\}$ and do grid search on each method, reporting their best score for each metric. Following the training phase, we evaluate the trained autoencoders on the test set. Similar to previous experiments, we use the metrics F ($F_{\text{latent}}$) and W ($W_{\text{latent}}$) in the latent space distributions $f_\phi \sharp \mu_1, \ldots, f_\phi \sharp \mu_K$ and the barycenter $\mu_0$. We use the reconstruction loss (binary cross-entropy, denoted as RL), the Wasserstein-2 distance between the prior and aggregated posterior distribution in latent space $W_{2,\text{latent}}^2 := W_2^2 \left( \mu_0, \frac{1}{K} \sum_{k=1}^K f_\phi \sharp \mu_k \right)$, as well as in image space $W_{2,\text{image}}^2 := W_2^2 \left( g_\psi \sharp \mu_0, \frac{1}{K} \sum_{k=1}^K \mu_k \right)$. Furthermore, we quantify the practical effect of the method by measuring Fairness metric in Image space. During evaluation, we approximate $\mu_0$ by its empirical version of 10000 samples. We report the quantitative result of grid search in Table 2, and reconstructed images, generated images, and images of latent codes in Figure 15 in Appendix D. From the results, the proposed surrogate MFSWB generally yield better scores than USWB, except for the generative score i.e, $W_{2,\text{image}}^2$. The formal MFSWB performs well in reconstruction loss and $W_{2,\text{image}}^2$, though its F and W scores are high. The $W_{2,\text{latent}}^2$ varies slightly across runs, with minor differences in performance order, indicating relatively similar results. While us-MFSWB achieves the best $F_{\text{images}}$ score, indicating the best fairness performance in image space, es-MFSWB excels in fairness within the latent space. Compared to conventional SWAE, using a barycenter loss results in a more class-fair latent representation but sacrifices image reconstruction and generative quality.

## 5 CONCLUSION

We introduced marginal fairness sliced Wasserstein barycenter (MFSWB), a special case of sliced Wasserstein barycenter (SWB) which has approximately the same distance to marginals. We first defined the MFSWB as a constrained uniform SWB problem. After that, to overcome the computational drawbacks of the original problem, we propose three surrogate definitions of MFSWB which are hyperparameter-free and easy to compute. We discussed the relationship of the proposed surrogate problems and their connection to the sliced Multi-marginal Wasserstein distance with the maximal ground metric. Finally, we conduct simulations with Gaussian and experiments on 3D point-cloud averaging, color harmonization, and sliced Wasserstein autoencoder with class-fairness representation to show the benefits of the proposed surrogate MFSWB definitions. Future works will focus on replacing SW with other metrics such as generalized sliced Wasserstein (Kolouri et al., 2019) and augmented sliced Wasserstein (Chen et al., 2022).

## ACKNOWLEDGEMENTS

We would like to thank Joydeep Ghosh for his insightful discussion during the course of this project.

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

# Supplement to "Marginal Fairness Sliced Wasserstein Barycenter"

We present skipped proofs in Appendix A. We then provide some additional materials which are mentioned in the main paper in Appendix B. After that, related works are discussed in Appendix C. We then provide additional experimental results in Appendix D. Finally, we report the used computational devices in Appendix E.

## A PROOFS

### A.1 PROOF OF PROPOSITION 1

*Proof.* From Definition 2, we have

$$\mathcal{SF}(\mu, \mu_{1:K}) = \max_{k \in \{1,\dots,K\}} SW_p^p(\mu, \mu_k)$$

$$= \max_{k \in \{1,\dots,K\}} \mathbb{E}_{\theta \sim \mathcal{U}(\mathbb{S}^{d-1})}[W_p^p(\theta\sharp\mu, \theta\sharp\mu_k)]$$

Let $k^\star = \arg\max_{k \in \{1,\dots,K\}} \mathbb{E}_{\theta \sim \mathcal{U}(\mathbb{S}^{d-1})}[W_p^p(\theta\sharp\mu, \theta\sharp\mu_k)]$, we have

$$\mathcal{SF}(\mu, \mu_{1:K}) = \mathbb{E}_{\theta \sim \mathcal{U}(\mathbb{S}^{d-1})}[W_p^p(\theta\sharp\mu, \theta\sharp\mu_{k^\star})]$$

$$\leq \mathbb{E}_{\theta \sim \mathcal{U}(\mathbb{S}^{d-1})} \left[ \max_{k \in \{1,\dots,K\}} W_p^p(\theta\sharp\mu, \theta\sharp\mu_k) \right]$$

$$= \mathcal{USF}(\mu, \mu_{1:K}),$$

as from Definition 3, which completes the proof.

### A.2 PROOF OF PROPOSITION 2

Using the Holder's inequality, we have:

$$\mathbb{E} \left| \nabla_\phi \frac{1}{L} \sum_{l=1}^{L} \mathbf{W}_p^p(\theta_l\sharp\mu_\phi, \theta_l\sharp\mu_{k_{\theta_l}^\star}) - \nabla_\phi \mathcal{USF}(\mu_\phi; \mu_{1:K}) \right|$$

$$\leq \left( \mathbb{E} \left| \nabla_\phi \frac{1}{L} \sum_{l=1}^{L} \mathbf{W}_p^p(\theta_l\sharp\mu_\phi, \theta_l\sharp\mu_{k_{\theta_l}^\star}) - \nabla_\phi \mathcal{USF}(\mu_\phi; \mu_{1:K}) \right|^2 \right)^{\frac{1}{2}}$$

$$= \left( \mathbb{E} \left( \nabla_\phi \frac{1}{L} \sum_{l=1}^{L} \mathbf{W}_p^p(\theta_l\sharp\mu_\phi, \theta_l\sharp\mu_{k_{\theta_l}^\star}) - \nabla_\phi \mathbb{E} \left[ \mathbf{W}_p^p(\theta\sharp\mu_\phi, \theta\sharp\mu_{k_\theta^\star}) \right] \right)^2 \right)^{\frac{1}{2}}$$

$$= \left( \mathbb{E} \left( \frac{1}{L} \sum_{l=1}^{L} \nabla_\phi \mathbf{W}_p^p(\theta_l\sharp\mu_\phi, \theta_l\sharp\mu_{k_{\theta_l}^\star}) - \mathbb{E} \left[ \nabla_\phi \mathbf{W}_p^p(\theta\sharp\mu_\phi, \theta\sharp\mu_{k_\theta^\star}) \right] \right)^2 \right)^{\frac{1}{2}}$$

$$= \left( \mathrm{Var} \left[ \frac{1}{L} \sum_{l=1}^{L} \nabla_\phi \mathbf{W}_p^p(\theta_l\sharp\mu_\phi, \theta_l\sharp\mu_{k_{\theta_l}^\star}) \right] \right)^{\frac{1}{2}}$$

$$= \frac{1}{\sqrt{L}} \mathrm{Var} \left[ \nabla_\phi \mathbf{W}_p^p(\theta\sharp\mu_\phi, \theta\sharp\mu_{k_\theta^\star}) \right]^{\frac{1}{2}},$$

which completes the proof.

### A.3 PROOF OF PROPOSITION 3

We first restate the following Lemma from (Nguyen et al., 2024b) and provide the proof for completeness.

**Lemma 1.** *For any $L \geq 1$, $0 \leq a_1 \leq a_2 \leq \ldots \leq a_L$ and $0 < b_1 \leq b_2 \leq \ldots \leq b_L$, we have:*

$$\frac{1}{L}(\sum_{i=1}^{L} a_i)(\sum_{i=1}^{L} b_i) \leq \sum_{i=1}^{L} a_i b_i. \tag{15}$$

*Proof.* For $L = 1$, we directly have $a_i b_i = a_i b_i$. Assuming that for $L$ the inequality holds i.e., $\frac{1}{L}(\sum_{i=1}^{L} a_i)(\sum_{i=1}^{L} b_i) \leq \sum_{i=1}^{L} a_i b_i$ which is equivalent to $(\sum_{i=1}^{L} a_i)(\sum_{i=1}^{L} b_i) \leq L \sum_{i=1}^{L} a_i b_i$. Now, we show that $\frac{1}{L}(\sum_{i=1}^{L} a_i)(\sum_{i=1}^{L} b_i) \leq \sum_{i=1}^{L} a_i b_i$ i.e., the inequality holds for $L+1$. We have

$$(\sum_{i=1}^{L+1} a_i)(\sum_{i=1}^{L+1} b_i) = (\sum_{i=1}^{L} a_i)(\sum_{i=1}^{L} b_i) + (\sum_{i=1}^{L} a_i)b_{L+1} + (\sum_{i=1}^{L} b_i)a_{L+1} + a_{L+1}b_{L+1}$$

$$\leq L \sum_{i=1}^{L} a_i b_i + (\sum_{i=1}^{L} a_i)b_{L+1} + (\sum_{i=1}^{L} b_i)a_{L+1} + a_{L+1}b_{L+1}.$$

Since $a_{L+1}b_{L+1} + a_i b_i \geq a_{L+1}b_i + b_{L+1}a_i$ for all $1 \leq i \leq L$ by rearrangement inequality. By taking the sum of these inequalities over $i$ from 1 to $L$, we obtain:

$$(\sum_{i=1}^{L} a_i)b_{L+1} + (\sum_{i=1}^{L} b_i)a_{L+1} \leq \sum_{i=1}^{L} a_i b_i + La_{L+1}b_{L+1}.$$

Then, we have

$$(\sum_{i=1}^{L+1} a_i)(\sum_{i=1}^{L+1} b_i) \leq L \sum_{i=1}^{L} a_i b_i + (\sum_{i=1}^{L} a_i)b_{L+1} + (\sum_{i=1}^{L} b_i)a_{L+1} + a_{L+1}b_{L+1}$$

$$\leq L \sum_{i=1}^{L} a_i b_i + \sum_{i=1}^{L} a_i b_i + La_{L+1}b_{L+1} + a_{L+1}b_{L+1}$$

$$= (L+1)(\sum_{i=1}^{L+1} a_i b_i),$$

which completes the proof. $\qquad\square$

Now, we go back to the main inequality which is $\mathcal{USF}(\mu; \mu_{1:K}) \leq \mathcal{ESF}(\mu; \mu_{1:K})$. From Definition 5, we have:

$$\mathcal{ESF}(\mu; \mu_{1:K}) = \mathbb{E}_{\theta \sim \sigma(\theta; \mu, \mu_{1:K})} \left[ \max_{k \in \{1,\ldots,K\}} W_p^p(\theta\sharp\mu, \theta\sharp\mu_k) \right]$$

$$= \mathbb{E}_{\theta \sim \mathcal{U}(\mathbb{S}^{d-1})} \left[ \max_{k \in \{1,\ldots,K\}} W_p^p(\theta\sharp\mu, \theta\sharp\mu_k) \frac{f_\sigma(\theta; \mu, \mu_{1:K})}{\frac{\Gamma(d/2)}{2\pi^{d/2}}} \right],$$

where $f_\sigma(\theta; \mu, \mu_{1:K}) \propto \exp\left(\max_{k \in \{1,\ldots,K\}} W_p^p(\theta\sharp\mu, \theta\sharp\mu_k)\right)$. Now, we consider a Monte Carlo estimation of $\mathcal{ESF}(\mu; \mu_{1:K})$ by importance sampling:

$$\widehat{\mathcal{ESF}}(\mu; \mu_{1:K}, L) = \frac{1}{L} \sum_{l=1}^{L} \left[ \max_{k \in \{1,\ldots,K\}} W_p^p(\theta_l\sharp\mu, \theta_l\sharp\mu_k) \frac{\exp\left(\max_{k \in \{1,\ldots,K\}} W_p^p(\theta_l\sharp\mu, \theta_l\sharp\mu_k)\right)}{\sum_{i=1}^{L} \exp\left(\max_{k \in \{1,\ldots,K\}} W_p^p(\theta_i\sharp\mu, \theta_i\sharp\mu_k)\right)} \right],$$

where $\theta_1, \ldots, \theta_L \overset{i.i.d}{\sim} \mathcal{U}(\mathbb{S}^{d-1})$. Similarly, we consider a Monte Carlo estimation of $\mathcal{USF}(\mu; \mu_{1:K})$:

$$\widehat{\mathcal{USF}}(\mu; \mu_{1:K}, L) = \frac{1}{L} \sum_{l=1}^{L} \left[ \max_{k \in \{1,\ldots,K\}} W_p^p(\theta_l\sharp\mu, \theta_l\sharp\mu_k) \right],$$

for the same set of $\theta_1, \ldots, \theta_L$. Without losing generality, we assume that $\max_{k \in \{1,\ldots,K\}} W_p^p(\theta_1\sharp\mu, \theta_1\sharp\mu_k) \leq \ldots \leq \max_{k \in \{1,\ldots,K\}} W_p^p(\theta_L\sharp\mu, \theta_L\sharp\mu_k)$. Let

$\max_{k \in \{1,\ldots,K\}} W_p^p(\theta_i \sharp \mu, \theta_i \sharp \mu_k) = a_i$ and $\exp\left(\max_{k \in \{1,\ldots,K\}} W_p^p(\theta_i \sharp \mu, \theta_i \sharp \mu_k)\right) = b_i$, applying Lemma 1, we have:

$$\widehat{\mathcal{USF}}(\mu; \mu_{1:K}, L) \leq \widehat{\mathcal{ESF}}(\mu; \mu_{1:K}, L) \quad \forall L \geq 1.$$

By letting $L \to \infty$ and applying the law of large numbers, we obtain:

$$\mathcal{USF}(\mu; \mu_{1:K}) \leq \mathcal{ESF}(\mu; \mu_{1:K}),$$

which completes the proof.

### A.4 PROOF OF PROPOSITION 4

We first recall the definition of the SMW with the maximal ground metric:

$$SMW_p^p(\mu_1, \ldots, \mu_K; c) = \mathbb{E}\left[\inf_{\pi \in \Pi(\mu_1, \ldots, \mu_K)} \int \max_{i \in \{1, \ldots, K\}, j \in \{1, \ldots, K\}} |\theta^\top x_i - \theta^\top x_j|^p d\pi(x_1, \ldots, x_K)\right].$$

**Non-negativity.** Since $\max_{i \in \{1, \ldots, K\}, j \in \{1, \ldots, K\}} |\theta^\top x_i - \theta^\top x_j|^p \geq 0$ for any $x_1, \ldots, x_K$ and for any $\theta$, we can obtain the desired property $SMW_p^p(\mu_1, \ldots, \mu_K; c) \geq 0$ which implies $SMW_p(\mu_1, \ldots, \mu_K; c) \geq 0$.

**Marginal Exchangeability.** For any permutation $\sigma : [[K]] \to [[K]]$, we have:

$$SMW_p^p(\mu_1, \ldots, \mu_K; c) = \mathbb{E}\left[\inf_{\pi \in \Pi(\mu_1, \ldots, \mu_K)} \int \max_{i \in \{1, \ldots, K\}, j \in \{1, \ldots, K\}} |\theta^\top x_i - \theta^\top x_j|^p d\pi(x_1, \ldots, x_K)\right]$$

$$= \mathbb{E}\left[\inf_{\pi \in \Pi(\mu_{\sigma(1)}, \ldots, \mu_{\sigma(K)})} \int \max_{i \in \{1, \ldots, K\}, j \in \{1, \ldots, K\}} |\theta^\top x_i - \theta^\top x_j|^p d\pi(x_1, \ldots, x_K)\right]$$

$$= SMW_p^p(\mu_{\sigma(1)}, \ldots, \mu_{\sigma(K)}; c).$$

**Generalized Triangle Inequality.** For $\mu \in \mathcal{P}_p(\mathbb{R}^d)$, we have :

$$SMW_p^p(\mu_1, \ldots, \mu_K; c)$$

$$= \mathbb{E}\left[\inf_{\pi \in \Pi(\mu_1, \ldots, \mu_K)} \int \max_{i \in \{1, \ldots, K\}, j \in \{1, \ldots, K\}} |\theta^\top x_i - \theta^\top x_j|^p d\pi(x_1, \ldots, x_K)\right]$$

$$\leq \mathbb{E}\left[\inf_{\pi \in \Pi(\mu_1, \ldots, \mu_K)} \int \sum_{k=1}^{K} \max_{i \in \{1, \ldots, K\} \setminus \{k\}, j \in \{1, \ldots, K\} \setminus \{k\}} |\theta^\top x_i - \theta^\top x_j|^p d\pi(x_1, \ldots, x_K)\right]$$

$$= \mathbb{E}\left[\inf_{\pi \in \Pi(\mu_1, \ldots, \mu_K)} \sum_{k=1}^{K} \int \max_{i \in \{1, \ldots, K\} \setminus \{k\}, j \in \{1, \ldots, K\} \setminus \{k\}} |\theta^\top x_i - \theta^\top x_j|^p d\pi(x_1, \ldots, x_K)\right]$$

$$= \mathbb{E}\left[\sum_{k=1}^{K} \int \max_{i \in \{1, \ldots, K\} \setminus \{k\}, j \in \{1, \ldots, K\} \setminus \{k\}} |\theta^\top x_i - \theta^\top x_j|^p d\pi^\star(x_1, \ldots, x_{k-1}, x_{k+1}, \ldots x_K)\right]$$

for $\pi^\star$ is the optimal multi-marginal transportation plan and $\pi^\star(x_1, \ldots, x_{k-1}, x_{k+1}, x_K)$ is the marginal joint distribution by integrating out $x_k$. By the gluing lemma (Peyré & Cuturi, 2020), there exists optimal plans $\pi^\star(x_1, \ldots, x_{k-1}, y, x_{k+1}, x_K)$ for any $k \in [[K]]$ and $y$ follows $\mu$. We further

have:

$$SMW_p^p(\mu_1, \ldots, \mu_K; c)$$

$$\leq \mathbb{E}\left[\sum_{k=1}^{K} \int \max\left(\max_{i \in \{1,\ldots,K\}\setminus\{k\}, j \in \{1,\ldots,K\}\setminus\{k\}} |\theta^\top x_i - \theta^\top x_j|^p, \right.\right.$$

$$\left.\left. \max_{i \in \{1,\ldots,K\}\setminus\{k\}} |\theta^\top x_i - \theta^\top y|^p \right) d\pi^\star(x_1, \ldots, x_{k-1}, y, x_{k+1}, \ldots x_K)\right]$$

$$= \sum_{k=1}^{K} \mathbb{E}\left[\inf_{\pi \in \Pi(\mu_1,\ldots,\mu_{k-1},\mu,\mu_{k+1},\ldots,\mu_K)} \int \max_{i \in \{1,\ldots,K\}, j \in \{1,\ldots,K\}} |\theta^\top x_i - \theta^\top x_j|^p d\pi(x_1, \ldots, x_K)\right]$$

$$= \sum_{k=1}^{K} SMW_p^p(\mu_1, \ldots, \mu_{k-1}, \mu, \mu_{k+1}, \ldots, \mu_K; c).$$

Applying the Minkowski's inequality, we obtain the desired property:

$$SMW_p(\mu_1, \ldots, \mu_K; c) \leq \sum_{k=1}^{K} SMW_p(\mu_1, \ldots, \mu_{k-1}, \mu, \mu_{k+1}, \ldots, \mu_K; c).$$

**Identity of Indiscernibles.** From the proof in Appendix A.5, we have:

$$SMW_p^p(\mu_1, \ldots, \mu_K; c) \geq \mathbb{E}\left[\max_{i \in \{1,\ldots,K\}, j \in \{1,\ldots,K\}} W_p^p(\theta \sharp \mu_i, \theta \sharp \mu_j)\right]$$

$$\geq \max_{i \in \{1,\ldots,K\}, j \in \{1,\ldots,K\}} \mathbb{E}\left[W_p^p(\theta \sharp \mu_i, \theta \sharp \mu_j)\right]$$

$$= \max_{i \in \{1,\ldots,K\}, j \in \{1,\ldots,K\}} SW_p^p(\mu_i, \mu_j).$$

Therefore, when $SMW_p(\mu_1, \ldots, \mu_K; c) = 0$, we have $SW_p^p(\mu_i, \mu_j) = 0$ which implies $\mu_i = \mu_j$ for any $i, j \in [[K]]$. As a result, $\mu_1 = \ldots = \mu_K$ from the metricity of the SW distance. For the other direction, it is easy to see that if $\mu_1 = \ldots \mu_K$, we have $SMW_p(\mu_1, \ldots, \mu_K; c) = 0$ based on the definition and the metricity of the Wasserstein distance.

## A.5 Proof of Proposition 5

Given the maximal ground metric $c(\theta^\top x_1, \ldots, \theta^\top x_K) = \max_{i \in \{1,\ldots,K\}, j \in \{1,\ldots,K\}} |\theta^\top x_i - \theta^\top x_j|$, from Equation 6

$$SMW_p^p(\mu_1, \ldots, \mu_K; c) = \mathbb{E}\left[\inf_{\pi \in \Pi(\mu_1,\ldots,\mu_K)} \int c(\theta^\top x_1, \ldots, \theta^\top x_K)^p d\pi(x_1, \ldots, x_K)\right]$$

$$= \mathbb{E}\left[\inf_{\pi \in \Pi(\mu_1,\ldots,\mu_K)} \int \max_{i \in \{1,\ldots,K\}, j \in \{1,\ldots,K\}} |\theta^\top x_i - \theta^\top x_j|^p d\pi(x_1, \ldots, x_K)\right]$$

By Jensen inequality i.e., $(x_1, \ldots, x_K) \to \max_{i \in \{1,\ldots,K\}, j \in \{1,\ldots,K\}} |\theta^\top x_i - \theta^\top x_j|^p$ is a convex function, we have:

$$SMW_p^p(\mu_1, \ldots, \mu_K; c) \geq \mathbb{E}\left[\inf_{\pi \in \Pi(\mu_1,\ldots,\mu_K)} \max_{i \in \{1,\ldots,K\}, j \in \{1,\ldots,K\}} \int |\theta^\top x_i - \theta^\top x_j|^p d\pi(x_1, \ldots, x_K)\right].$$

Using max-min inequality, we have:

$$SMW_p^p(\mu_1, \ldots, \mu_K; c) \geq \mathbb{E}\left[\max_{i \in \{1,\ldots,K\}, j \in \{1,\ldots,K\}} \inf_{\pi \in \Pi(\mu_1,\ldots,\mu_K)} \int |\theta^\top x_i - \theta^\top x_j|^p d\pi(x_1, \ldots, x_K)\right]$$

$$\geq \mathbb{E}\left[\max_{i \in \{1,\ldots,K\}, j \in \{1,\ldots,K\}} \inf_{\pi \in \Pi(\mu_i,\mu_j)} \int |\theta^\top x_i - \theta^\top x_j|^p d\pi(x_i, x_j)\right]$$

$$= \mathbb{E}\left[\max_{i \in \{1,\ldots,K\}, j \in \{1,\ldots,K\}} W_p^p(\theta \sharp \mu_i, \theta \sharp \mu_j)\right].$$

---

**Algorithm 1** Computational algorithm of the SWB problem

---

**Input:** Marginals $\mu_1, \ldots, \mu_K$, $p \geq 1$, weights $\omega_1, \ldots, \omega_K$, the number of projections $L$, step size $\eta$, the number of iterations $T$.
Initialize the barycenter $\mu_\phi$
**for** $t = 1$ to $T$ **do**
    Set $\nabla_\phi = 0$
    Sample $\theta_l \sim \mathcal{U}(\mathbb{S}^{d-1})$
    **for** $l = 1$ to $L$ **do**
        **for** $k = 1$ to $K$ **do**
            Set $\nabla_\phi = \nabla_\phi + \nabla_\phi \frac{\omega_k}{L} \mathrm{W}_p^p(\theta_l \sharp \mu_\phi, \theta_l \sharp \mu_k)$
        **end for**
    **end for**
    $\phi = \phi - \eta \nabla_\phi$
**end for**
**Return:** $\mu_\phi$

---

**Algorithm 2** Computational algorithm of the s-MFSWB problem

---

**Input:** Marginals $\mu_1, \ldots, \mu_K$, $p \geq 1$ the number of projections $L$, step size $\eta$, the number of iterations $T$.
Initialize the barycenter $\mu_\phi$
**for** $t = 1$ to $T$ **do**
    Set $\nabla_\phi = 0$
    Sample $\theta_l \sim \mathcal{U}(\mathbb{S}^{d-1})$
    $k^\star = 1$
    **for** $k = 1$ to $K$ **do**
        **for** $l = 1$ to $L$ **do**
            **if** $\frac{1}{L} \sum_{l=1}^L \mathrm{W}_p^p(\theta_l \sharp \mu_\phi, \theta_l \sharp \mu_k) > \frac{1}{L} \sum_{l=1}^L \mathrm{W}_p^p(\theta_l \sharp \mu_\phi, \theta_l \sharp \mu_{k^\star})$ **then**
            $k^\star = k$
            **end if**
        **end for**
    **end for**
    $\nabla_\phi = \nabla_\phi + \frac{1}{L} \sum_{l=1}^L \nabla_\phi \mathrm{W}_p^p(\theta_l \sharp \mu_\phi, \theta_l \sharp \mu_{k^\star})$
    $\phi = \phi - \eta \nabla_\phi$
**end for**
**Return:** $\mu_\phi$

---

Therefore, minimizing two sides with respect to $\mu_1$, we have:

$$\min_{\mu_1} SMW_p^p(\mu_1, \ldots, \mu_K; c) \geq \min_{\mu_1} \mathbb{E}\left[\max_{i \in \{1,\ldots,K\}, j \in \{1,\ldots,K\}} W_p^p(\theta \sharp \mu_i, \theta \sharp \mu_j)\right]$$

$$\geq \min_{\mu_1} \mathbb{E}\left[\max_{i \in \{2,\ldots,K\}} W_p^p(\theta \sharp \mu_1, \theta \sharp \mu_i)\right]$$

$$= \min_{\mu_1} \mathcal{USF}(\mu_1; \mu_{2:K}),$$

which completes the proof.

## B   ADDITIONAL MATERIALS

**Algorithms.** As mentioned in the main paper, we present the computational algorithm for SWB in Algorithm 1, for s-MFSWB in Algorithm 2, for us-MFSWB in Algorithm 3, and for es-MFSWB in Algorithm 4.

**Energy-based Sliced Multi-marginal Wasserstein.** As shown in Proposition 5, us-MFSWB is equivalent to minimizing a lower bound of SMW with the maximal ground metric. We now show that es-MFSWB is also equivalent to minimizing a lower bound of a variant of SMW i.e., Energy-based sliced Multi-marginal Wasserstein with the maximal ground metric. We refer the reader

---

**Algorithm 3** Computational algorithm of the us-MFSWB problem

---

**Input:** Marginals $\mu_1, \ldots, \mu_K$, $p \geq 1$ the number of projections $L$, step size $\eta$, the number of iterations $T$.

Initialize the barycenter $\mu_\phi$

**for** $t = 1$ to $T$ **do**

    Set $\nabla_\phi = 0$

    Sample $\theta_l \sim \mathcal{U}(\mathbb{S}^{d-1})$

    **for** $l = 1$ to $L$ **do**

        $k_l^\star = 1$

        **for** $k = 2$ to $K$ **do**

            **if** $\mathrm{W}_p^p(\theta_l \sharp \mu_\phi, \theta_l \sharp \mu_k) > \mathrm{W}_p^p(\theta_l \sharp \mu_\phi, \theta_l \sharp \mu_{k_l^\star})$ **then**

                $k_l^\star = k$

            **end if**

        **end for**

        $\nabla_\phi = \nabla_\phi + \nabla_\phi \frac{1}{L} \mathrm{W}_p^p(\theta_l \sharp \mu_\phi, \theta_l \sharp \mu_{k_l^\star})$

    **end for**

    $\phi = \phi - \eta \nabla_\phi$

**end for**

**Return:** $\mu_\phi$

---

**Algorithm 4** Computational algorithm of the es-MFSWB problem

---

**Input:** Marginals $\mu_1, \ldots, \mu_K$, $p \geq 1$ the number of projections $L$, step size $\eta$, the number of iterations $T$.

Initialize the barycenter $\mu_\phi$

**for** $t = 1$ to $T$ **do**

    Set $\nabla_\phi = 0$

    Sample $\theta_l \sim \mathcal{U}(\mathbb{S}^{d-1})$

    **for** $l = 1$ to $L$ **do**

        $k_l^\star = 1$

        **for** $k = 2$ to $K$ **do**

            **if** $\mathrm{W}_p^p(\theta_l \sharp \mu_\phi, \theta_l \sharp \mu_k) > \mathrm{W}_p^p(\theta_l \sharp \mu_\phi, \theta_l \sharp \mu_{k_l^\star})$ **then**

                $k_l^\star = k$

            **end if**

        **end for**

    **end for**

    **for** $l = 1$ to $L$ **do**

        $w_{l,\phi} = \frac{\exp(\mathrm{W}_p^p(\theta_l \sharp \mu_\phi, \theta_l \sharp \mu_{k_l^\star}))}{\sum_{j=1}^L \exp(\mathrm{W}_p^p(\theta_j \sharp \mu_\phi, \theta_j \sharp \mu_{k_j^\star}))}$

    **end for**

    $\nabla_\phi = \nabla_\phi + \nabla_\phi \frac{w_{l,\phi}}{L} \mathrm{W}_p^p(\theta_l \sharp \mu_\phi, \theta_l \sharp \mu_{k_l^\star})$

    $\phi = \phi - \eta \nabla_\phi$

**end for**

**Return:** $\mu_\phi$

---

to Proposition 6 for a detailed definition. The proof of Proposition 6 is similar to the proof of Proposition 5 in Appendix A.5.

**Proposition 6.** *Given* $K \geq 2$ *marginals* $\mu_1, \ldots, \mu_K \in \mathcal{P}_p(\mathbb{R}^d)$, *the maximal ground metric* $c(\theta^\top x_1, \ldots, \theta^\top x_K) = \max_{i \in \{1, \ldots, K\}, j \in \{1, \ldots, K\}} |\theta^\top x_i - \theta^\top x_j|$, *we have:*

$$\min_{\mu_1} \mathcal{ESF}(\mu_1; \mu_{2:K}) \leq \min_{\mu_1} ESMW_p^p(\mu_1, \mu_2, \ldots, \mu_K; c), \tag{16}$$

*where*

$$ESMW_p^p(\mu_1, \mu_2, \ldots, \mu_K; c) = \mathbb{E}\left[ \inf_{\pi \in \Pi(\mu_1, \ldots, \mu_K)} \int c(\theta^\top x_1, \ldots, \theta^\top x_K)^p d\pi(x_1, \ldots, x_K) \right],$$

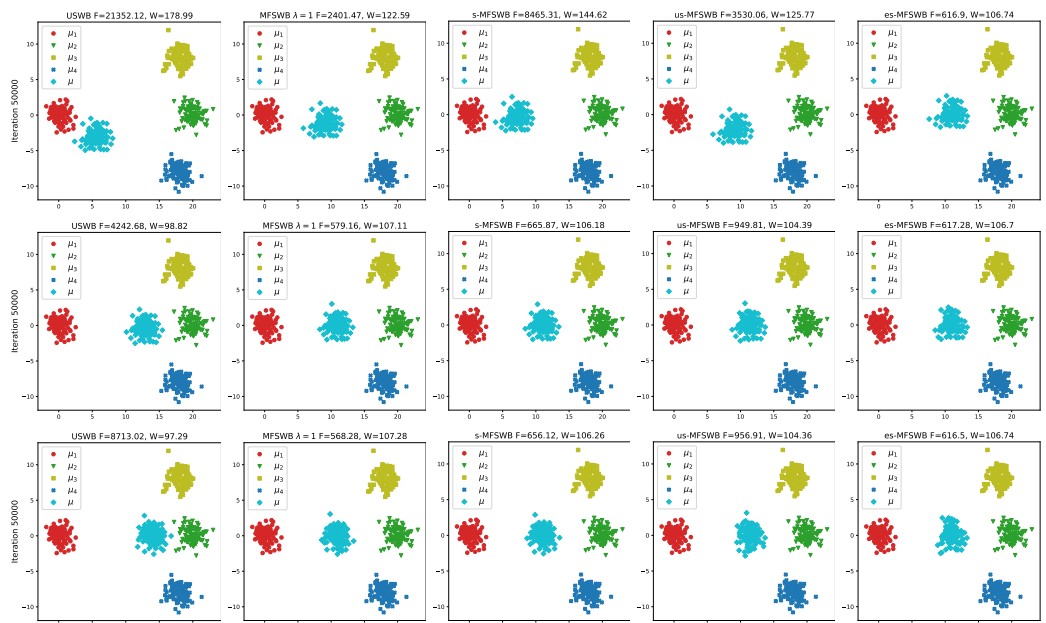

Figure 5: Barycenters from USWB, MFSWB with $\lambda = 1$, s-MFSWB, us-MFSWB, and es-MFSWB with learning rate 0.001 (first row), 0.005 (second row), and 0.05 (third row).

*and the expectation is with respect to $\sigma(\theta)$ i.e.,*

$$f_\sigma(\theta; \mu_1, \mu_{2:K}) \propto \exp\left(\max_{k \in \{2,...,K\}} W_p^p(\theta \sharp \mu_1, \theta \sharp \mu_k)\right).$$

## C  RELATED WORKS

**Fair Learning with Wasserstein Barycenter.** A connection between fair regression and one-dimensional Wasserstein barycenter is established by deriving the expression for the optimal function minimizing squared risk under Demographic Parity constraints (Chzhen et al., 2020). Similarly, Demographic Parity fair classification is connected to one-dimensional Wasserstein-1 distance barycenter in (Jiang et al., 2020). The work (Hu et al., 2023) extends the Demographic Parity constraint to multi-task problems for regression and classification and connects them to the one-dimensional Wasserstein-2 distance barycenters. A method to augment the input so that predictability of the protected attribute is impossible, by using Wasserstein-2 distance Barycenters to repair the data is proposed in (Gordaliza et al., 2019). A general approach for using one-dimensional Wasserstein-1 distance barycenter to obtain Demographic Parity in classification and regression is proposed in (Silvia et al., 2020). Overall, all discussed works define fairness in terms of Demographic Parity constraints in applications with a response variable (classification and regression) in one dimension. In contrast, we focus on marginal fairness barycenter i.e., using a set of measures only, in any dimensions.

**Other possible applications.** Wasserstein barycenter has been used to cluster measures in (Zhuang et al., 2022). In particular, a K-mean algorithm for measures is proposed with Wasserstein barycenter as the averaging operator. Therefore, our MFSWB can be directly used to enforce the fairness for averaging inside each cluster. The proposed MFSWB can be also used to average meshes by changing the SW to H2SW which is proposed in (Nguyen & Ho, 2024).

## D  ADDITIONAL EXPERIMENTS

**Gaussians barycenter with the formal MFSWB.** We report the result of finding barycenters from USWB, MFSWB with $\lambda = 1$, s-MFSWB, us-MFSWB, and es-MFSWB with learning rate 0.001, 0.005, and 0.05 in Figure 5. We present the result of finding barycenters of Gaussian distributions with MFSWB $\lambda = 0.1$ and $\lambda = 10$ in Figure 6.

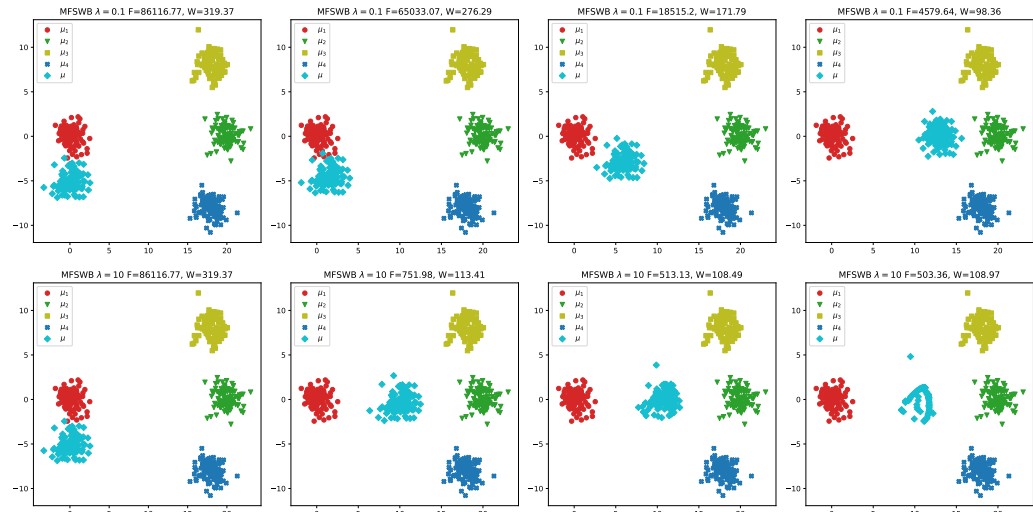

Figure 6: Barycenters from MFSWB with $\lambda = 0.1$ and $\lambda = 10$ along gradient iterations with the corresponding F-metric and W-metric.

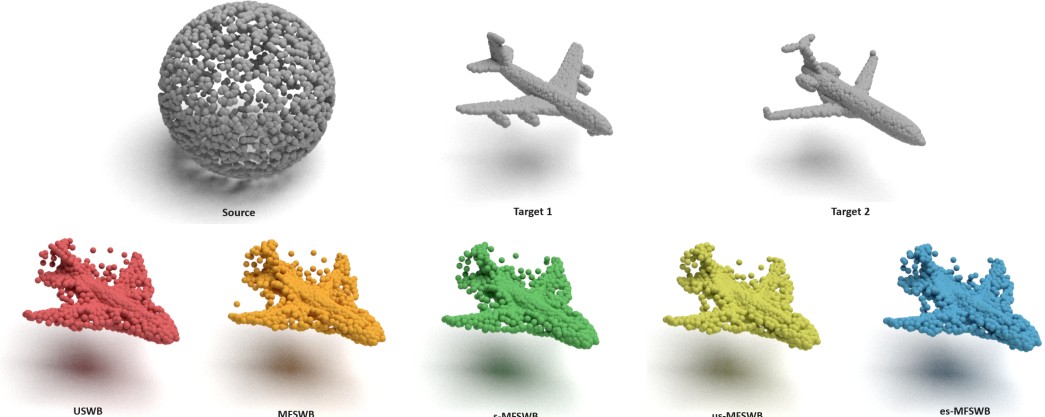

Figure 7: Averaging point-clouds with USWB, MFSWB ($\lambda = 1$), s-MFSWB, us-MFSWB, and es-MFSWB.

Table 3: F-metric and W-metric along iterations in point-cloud averaging application.

| Method | Iteration 0 | | Epoch 1000 | | Epoch 5000 | | Epoch 10000 | |
|---|---|---|---|---|---|---|---|---|
| | F ($\downarrow$) | W ($\downarrow$) | F ($\downarrow$) | W ($\downarrow$) | F ($\downarrow$) | W ($\downarrow$) | F ($\downarrow$) | W ($\downarrow$) |
| USWB | $746.67 \pm 0.0$ | $4814.71 \pm 0.0$ | $35.22 \pm 1.04$ | $161.11 \pm 0.54$ | $7.82 \pm 0.26$ | $109.82 \pm 0.28$ | $11.08 \pm 0.06$ | $108.52 \pm 0.17$ |
| MFSWB $\lambda = 0.1$ | $746.67 \pm 0.0$ | $4814.71 \pm 0.0$ | $35.15 \pm 0.36$ | $159.84 \pm 0.55$ | $4.95 \pm 0.23$ | $109.14 \pm 0.33$ | $6.95 \pm 0.8$ | $107.83 \pm 0.16$ |
| MFSWB $\lambda = 1$ | $746.67 \pm 0.0$ | $4814.71 \pm 0.0$ | $33.21 \pm 2.72$ | $151.24 \pm 0.64$ | $2.54 \pm 1.5$ | $109.66 \pm 0.26$ | $4.66 \pm 2.1$ | $\mathbf{108.1 \pm 0.05}$ |
| MFSWB $\lambda = 10$ | $746.67 \pm 0.0$ | $4814.71 \pm 0.0$ | $34.03 \pm 22.6$ | $158.66 \pm 1.39$ | $29.19 \pm 14.29$ | $122.66 \pm 0.88$ | $20.55 \pm 13.57$ | $123.65 \pm 1.52$ |
| s-MFSWB | $746.67 \pm 0.0$ | $4814.71 \pm 0.0$ | $36.23 \pm 1.88$ | $154.4 \pm 0.67$ | $\mathbf{0.66 \pm 0.44}$ | $109.17 \pm 0.34$ | $2.54 \pm 2.06$ | $107.57 \pm 0.19$ |
| us-MFSWB | $746.67 \pm 0.0$ | $4814.71 \pm 0.0$ | $28.65 \pm 1.37$ | $144.27 \pm 0.65$ | $1.02 \pm 0.8$ | $109.67 \pm 0.1$ | $\mathbf{1.35 \pm 0.77}$ | $108.2 \pm 0.19$ |
| es-MFSWB | $746.67 \pm 0.0$ | $4814.71 \pm 0.0$ | $\mathbf{28.05 \pm 1.16}$ | $\mathbf{143.24 \pm 0.76}$ | $0.99 \pm 0.32$ | $109.68 \pm 0.14$ | $1.36 \pm 0.62$ | $108.28 \pm 0.07$ |

**Point-cloud averaging.** We report the averaging results of two point-clouds of plane shapes n Figure 7 and the corresponding F-metrics and W-metric along iterations in Table 3. We see that the proposed surrogates achieve better F-metric and W-metric than the USWB. In this case, us-MFSWB gives the best F-metric at the final epoch, however, es-MFSWB also gives a comparable performance and performs better at earlier epochs. For the formal MFSWB, it does not perform well with the chosen set of $\lambda$.

**Color Harmonization.** We first present the harmonized images of different methods including USWB, MFSWB ($\lambda = 1$), s-MFSWB, us-MFSWB, and es-MFSWB at iteration 5000 and 10000 for the demonstrated images in the main text in Figure 8-Figure 9. Moreover, we report the results of MFSWB ($\lambda = 0.1, 10$) at iteration 5000, 10000, and 20000 in Figure 10. Similarly, we repeat the same experiments with flower images in Figure 11- 14. Overall, we see that es-MFSWB helps to

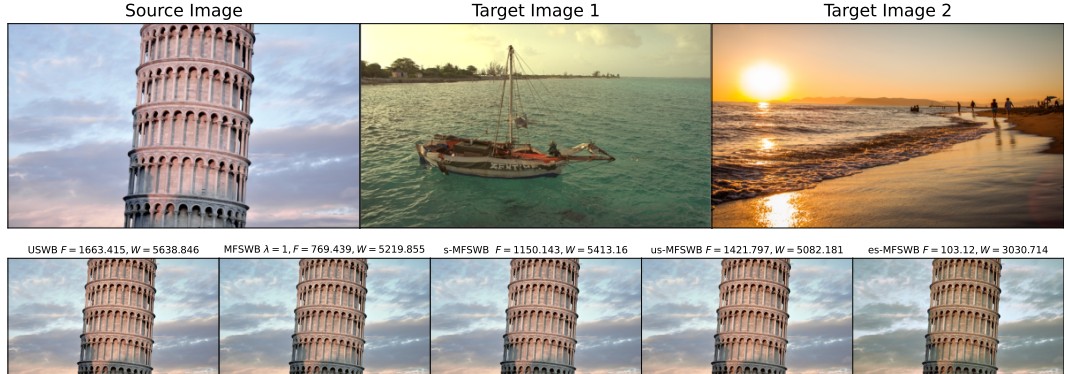

Figure 8: Harmonized images from USWB, MFSWB ($\lambda = 1$), s-MFSWB, us-MFSWB, and es-MFSWB at iteration 5000.

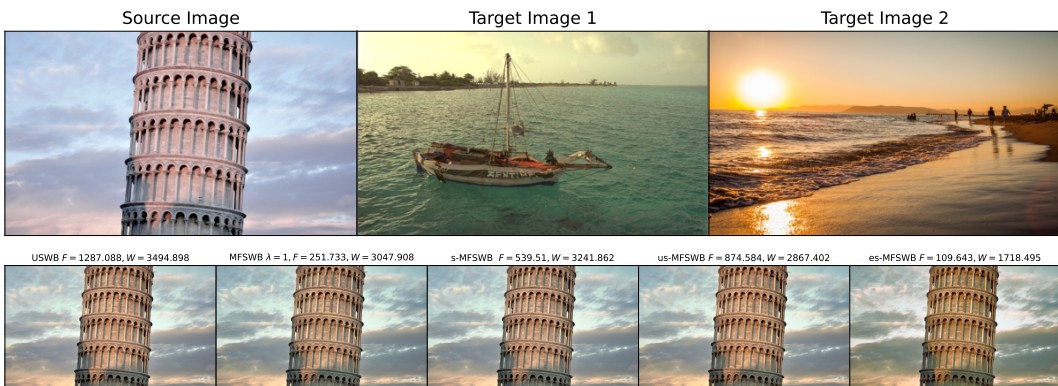

Figure 9: Harmonized images from USWB, MFSWB ($\lambda = 1$), s-MFSWB, us-MFSWB, and es-MFSWB at iteration 10000.

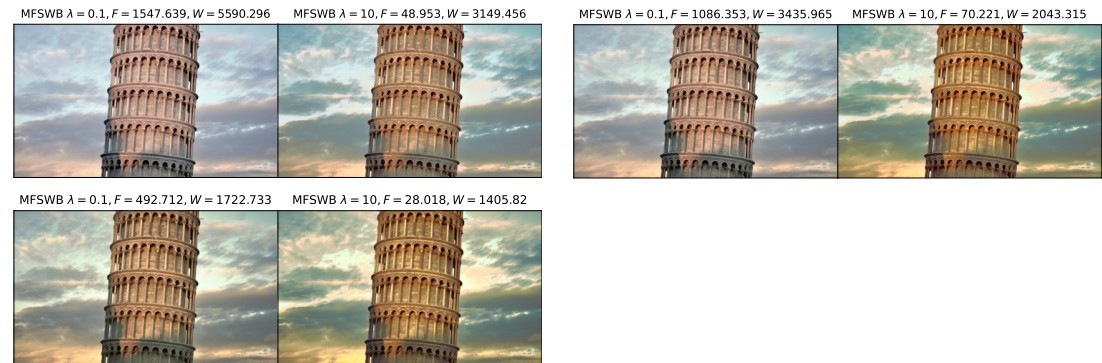

Figure 10: Harmonized images from MFSWB with $\lambda = 0.1$ and $\lambda = 10$ at iterations 5000, 10000, and 20000.

reduce both F-metric and W-metric faster than USWB and other surrogates. For the formal MFSWB, the performance depends significantly on the choice of $\lambda$.

**Sliced Wasserstein autoencoder with class-fairness representation.** We use the RMSprop optimizer with learning rate $0.01$, alpha=$0.99$, eps=$1e-8$. As mentioned in the main text, we report the used neural network architectures:

We report some randomly selected reconstructed images, some randomly generated images, and the test latent codes of trained autoencoders in Figure 15. Overall, we observe that the qualitative results are consistent with the quantitive results in Table 2. From the latent spaces, we see that the proposed

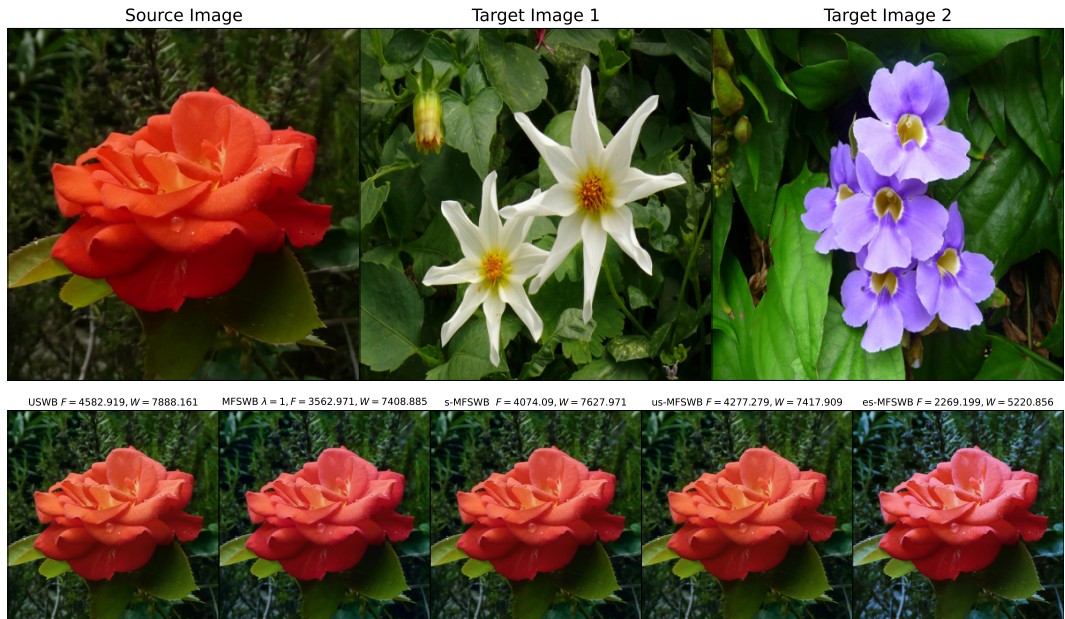

Figure 11: Harmonized images from USWB, MFSWB ($\lambda = 1$) s-MFSWB, us-MFSWB, and es-MFSWB at iteration 5000.

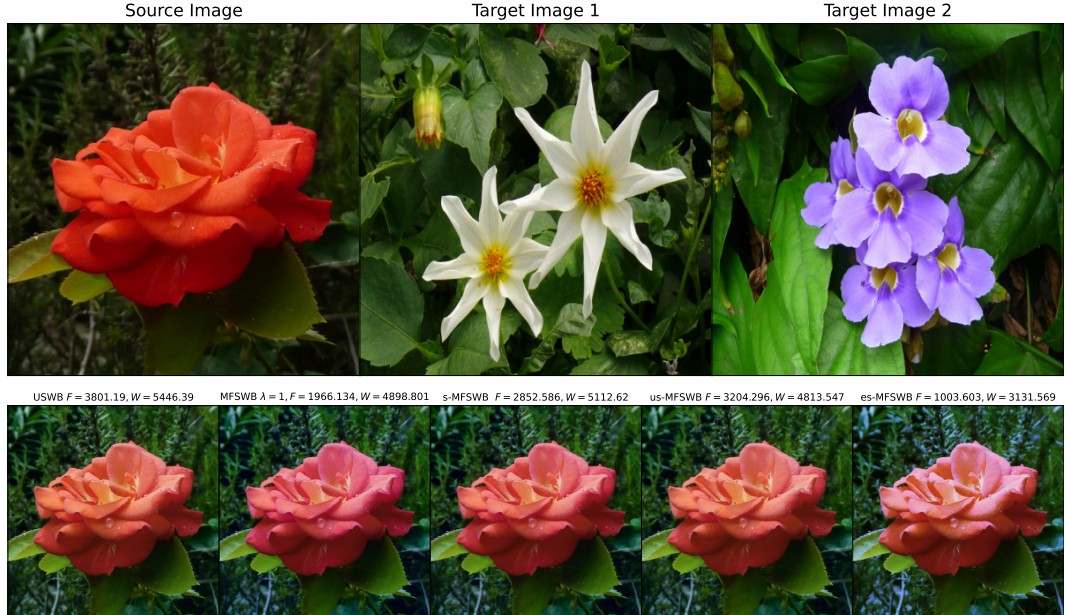

Figure 12: Harmonized images from USWB, MFSWB ($\lambda = 1$), s-MFSWB, us-MFSWB, and es-MFSWB at iteration 10000.

surrogates helps to make the codes of classes have approximately the same structure which do appear in the conventional SWAE's latent codes.

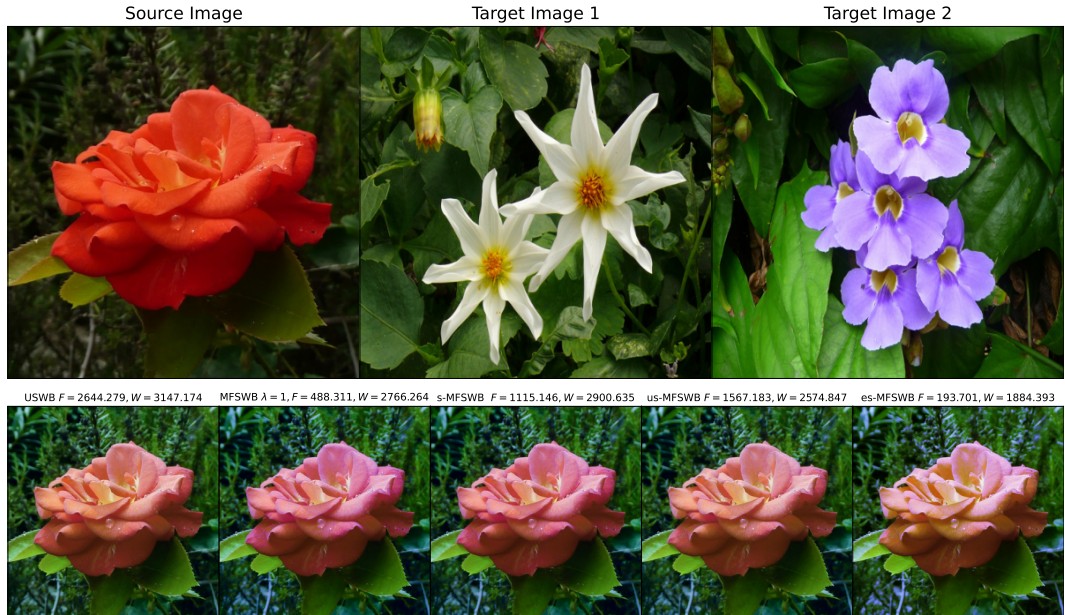

Figure 13: Harmonized images from USWB, MFSWB ($\lambda = 1$), s-MFSWB, us-MFSWB, and es-MFSWB at iterations 20000.

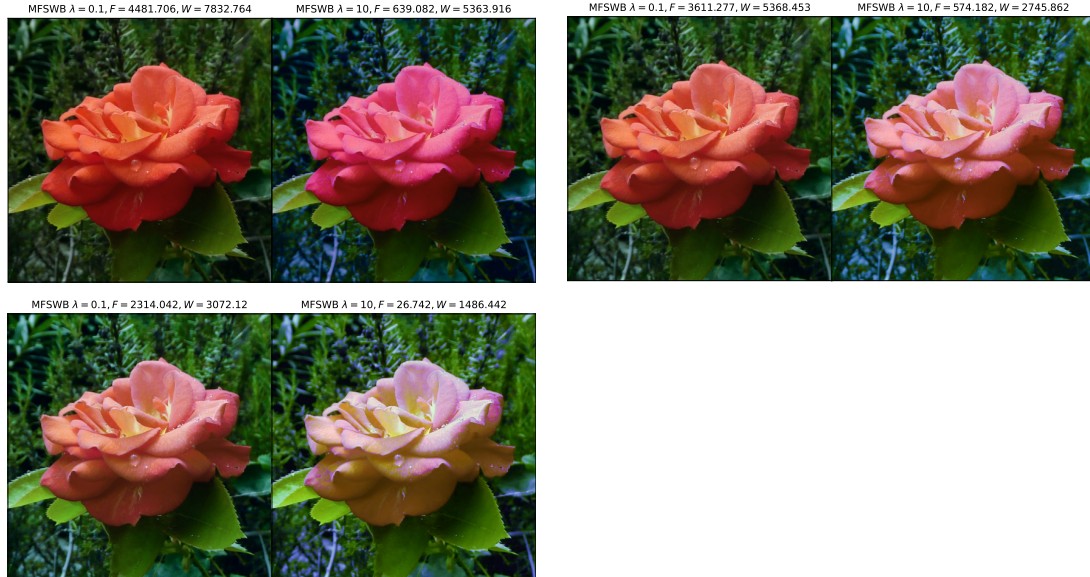

Figure 14: Color harmonized images from MFSWB with $\lambda = 0.1$ and $\lambda = 10$ at iterations 5000, 10000, and 20000.

# E    COMPUTATIONAL DEVICES

For the Gaussian simulation, point-cloud averaging, and color harmonization, we use a HP Omen 25L desktop for conducting experiments. Additionally, for the Sliced Wasserstein Autoencoder with class-fair representation experiment, we employ the NVIDIA Tesla V100 GPU.

| Method | Reconstructed Images | Generated Images | Latent Space |
|---|---|---|---|

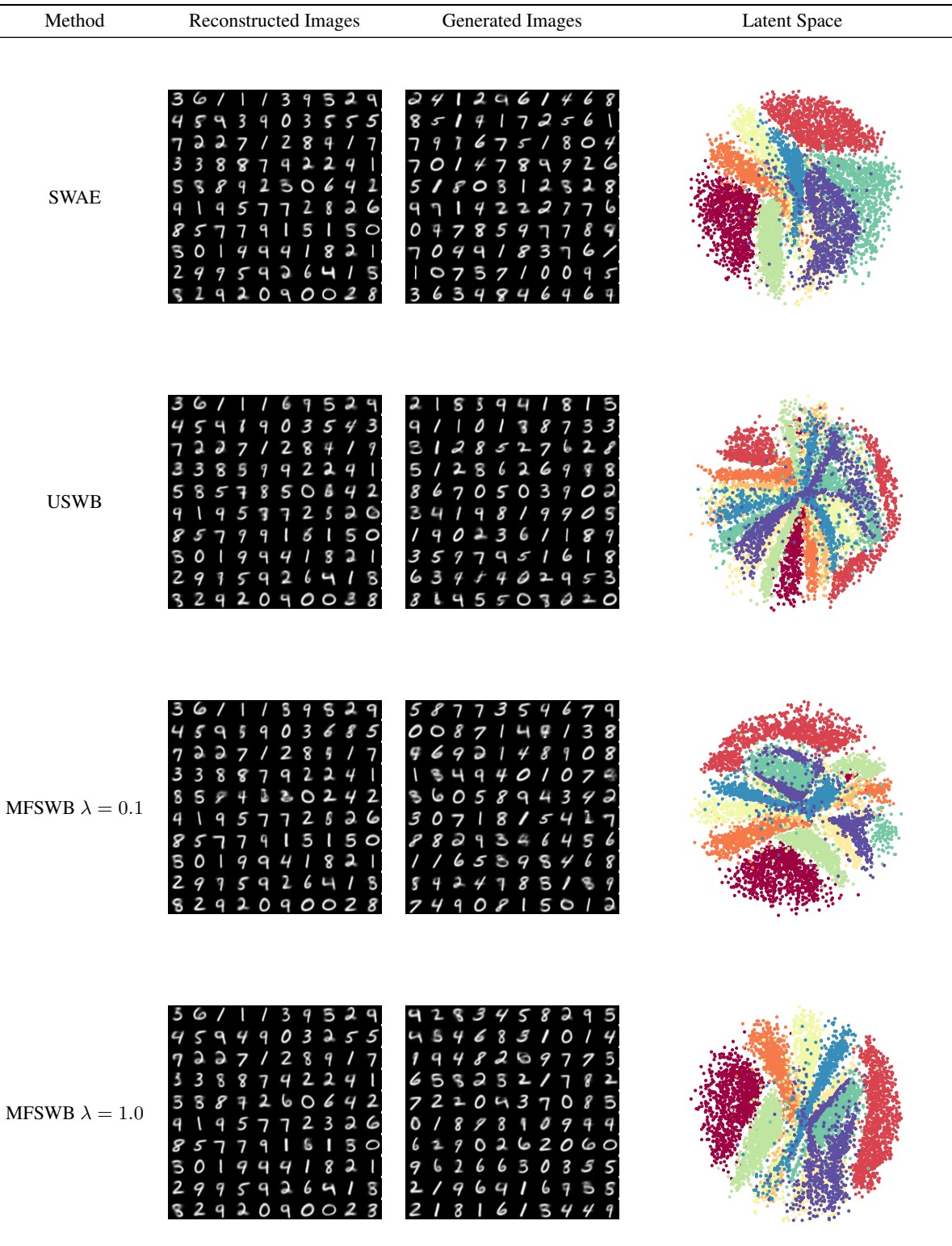

| Method | Reconstructed Images | Generated Images | Latent Space |
| --- | --- | --- | --- |

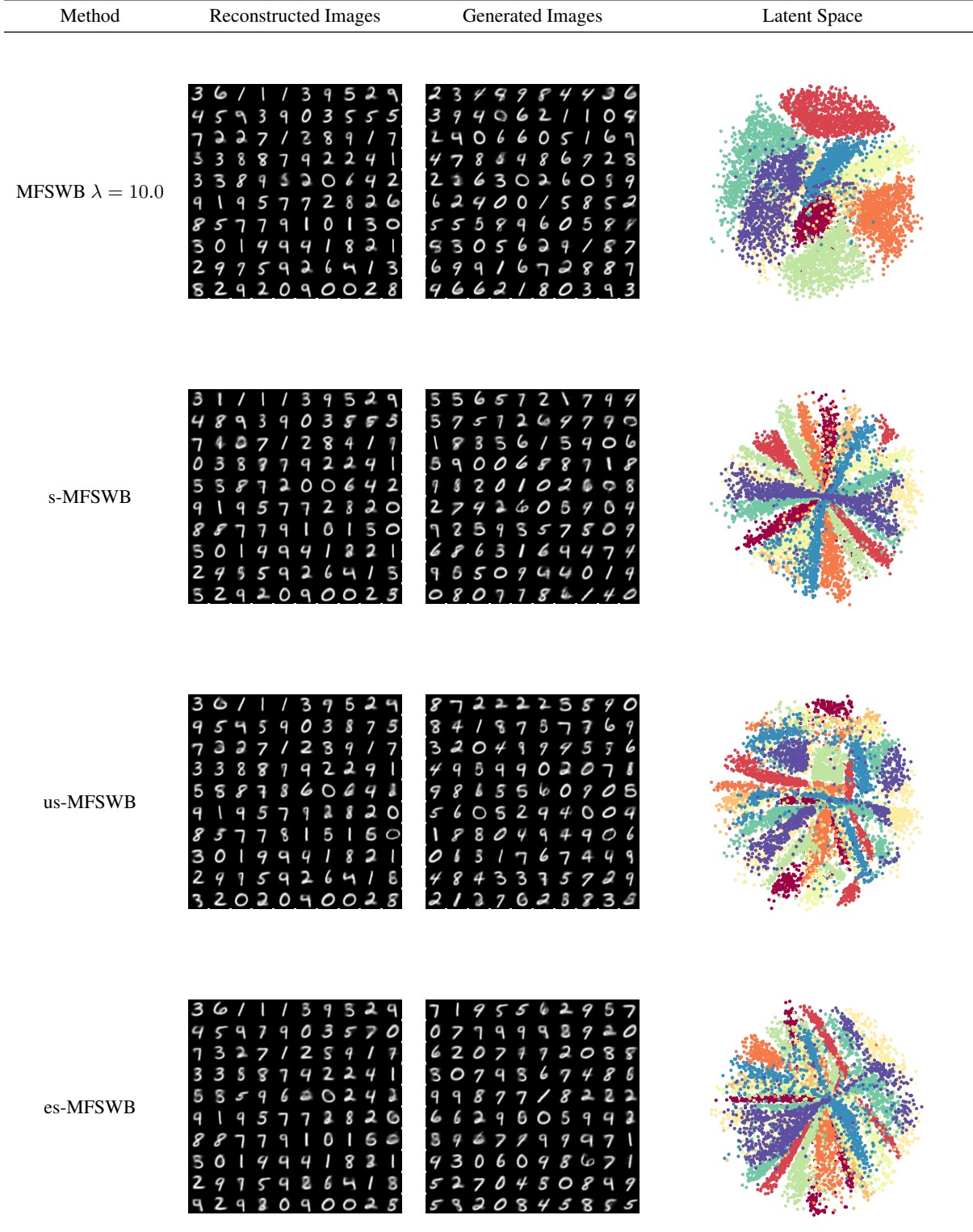

Figure 15: Reconstructed images, generated images and latent space of all methods.

| Layer | Description |
|---|---|
| **MNISTAutoencoder** | |
| **Encoder** | |
| Conv2d | (1, 16, kernel size=3, stride=1, padding=1) |
| LeakyReLU | (negative slope=0.2, inplace=True) |
| Conv2d | (16, 16, kernel size=3, stride=1, padding=1) |
| LeakyReLU | (negative slope=0.2, inplace=True) |
| AvgPool2d | (kernel size=2) |
| Conv2d | (16, 32, kernel size=3, stride=1, padding=1) |
| LeakyReLU | (negative slope=0.2, inplace=True) |
| Conv2d | (32, 32, kernel size=3, stride=1, padding=1) |
| LeakyReLU | (negative slope=0.2, inplace=True) |
| AvgPool2d | (kernel size=2) |
| Conv2d | (32, 64, kernel size=3, stride=1, padding=1) |
| LeakyReLU | (negative slope=0.2, inplace=True) |
| Conv2d | (64, 64, kernel size=3, stride=1, padding=1) |
| LeakyReLU | (negative slope=0.2, inplace=True) |
| AvgPool2d | (kernel size=2, padding=1) |
| Linear | (in_features=1024, out_features=128) |
| ReLU | (inplace=True) |
| Linear | (in_features=128, out_features=2) |
| **Decoder** | |
| Linear | (in_features=2, out_features=128) |
| Linear | (in_features=128, out_features=1024) |
| ReLU | (inplace=True) |
| Upsample | (scale_factor=2, mode=nearest) |
| Conv2d | (64, 64, kernel size=3, stride=1, padding=1) |
| LeakyReLU | (negative slope=0.2, inplace=True) |
| Conv2d | (64, 64, kernel size=3, stride=1, padding=1) |
| LeakyReLU | (negative slope=0.2, inplace=True) |
| Upsample | (scale_factor=2, mode=nearest) |
| Conv2d | (64, 64, kernel size=3, stride=1) |
| LeakyReLU | (negative slope=0.2, inplace=True) |
| Conv2d | (64, 64, kernel size=3, stride=1, padding=1) |
| LeakyReLU | (negative slope=0.2, inplace=True) |
| Upsample | (scale_factor=2, mode=nearest) |
| Conv2d | (64, 32, kernel size=3, stride=1, padding=1) |
| LeakyReLU | (negative slope=0.2, inplace=True) |
| Conv2d | (32, 32, kernel size=3, stride=1, padding=1) |
| LeakyReLU | (negative slope=0.2, inplace=True) |
| Conv2d | (32, 1, kernel size=3, stride=1, padding=1) |

Table 4: MNIST Autoencoder Architecture

Table 5: Comparison of methods with $\kappa_2 = 0.5$ on CIFAR10 after 500 epochs.

| Methods | RL ($\downarrow$) | $W^2_{2,\text{latent}}$ ($\downarrow$) | $W^2_{2,\text{image}}$ ($\downarrow$) | $F_{\text{latent}}$ ($\downarrow$) | $W_{\text{latent}}$ ($\downarrow$) | $F_{\text{images}}$ ($\downarrow$) | $W_{\text{images}}$ ($\downarrow$) |
|---|---|---|---|---|---|---|---|
| SWAE | 0.640 | 6.101 | 141.984 | 0.280 | 4.585 | 46.006 | 178.798 |
| UBSW | 0.640 | 6.104 | 135.944 | 0.228 | 4.572 | 44.024 | 174.322 |
| MFSWB $\lambda = 0.1$ | 0.640 | 6.097 | 142.530 | 0.281 | 4.585 | 46.080 | 179.210 |
| MFSWB $\lambda = 1.0$ | 0.641 | 6.092 | 142.289 | 0.279 | 4.578 | 46.076 | 179.135 |
| MFSWB $\lambda = 10.0$ | 0.640 | 6.100 | 141.503 | 0.282 | 4.585 | 46.088 | 178.373 |
| s-MFBSW | 0.640 | 6.103 | 134.766 | 0.218 | 4.569 | 42.503 | 173.530 |
| us-MFBSW | 0.642 | 6.088 | **131.934** | **0.209** | 4.546 | **39.329** | **171.204** |
| es-MFBSW | 0.642 | **6.060** | 132.170 | 0.212 | **4.534** | 40.642 | 171.573 |

Table 6: Comparison of methods with $\kappa_2 = 0.5$ on STL10 after 500 epochs.

| Methods | RL ($\downarrow$) | $W^2_{2,\text{latent}}$ ($\downarrow$) | $W^2_{2,\text{image}}$ ($\downarrow$) | $F_{\text{latent}}$ ($\downarrow$) | $W_{\text{latent}}$ ($\downarrow$) | $F_{\text{images}}$ ($\downarrow$) | $W_{\text{images}}$ ($\downarrow$) |
|---|---|---|---|---|---|---|---|
| SWAE | 0.613 | 16.826 | 301.397 | 0.647 | 15.699 | 25.827 | 199.175 |
| UBSW | 0.616 | 16.908 | 301.143 | 0.585 | 15.719 | 24.905 | 199.918 |
| MFSWB $\lambda = 0.1$ | 0.614 | 16.823 | 301.704 | 0.647 | 15.698 | 25.637 | 199.662 |
| MFSWB $\lambda = 1.0$ | 0.614 | 16.814 | 301.505 | 0.647 | 15.688 | 25.790 | 199.307 |
| MFSWB $\lambda = 10.0$ | 0.613 | 16.831 | 301.370 | 0.648 | 15.705 | 25.546 | 199.168 |
| s-MFBSW | 0.613 | 16.842 | 302.632 | 0.580 | 15.658 | 23.520 | 200.262 |
| us-MFBSW | 0.616 | 16.830 | 297.952 | 0.586 | **15.645** | 23.638 | **197.057** |
| es-MFBSW | 0.616 | **16.796** | **296.548** | **0.557** | 15.658 | **22.551** | 199.117 |

**Results.** We evaluate the scalability of our method using two well-established datasets: CIFAR10 (Krizhevsky et al., 2009) ($d = 32 \times 32 \times 3$) and STL10 (Coates et al., 2011) ($d = 64 \times 64 \times 3$). For these experiments, we set $\kappa_1 = 8.0$, $\kappa_2 = 0.5$, and train for 500 epochs with a learning rate of 0.0005. The CIFAR10 experiment uses a uniform distribution on a 48-dimensional ball ($h = 48$), while the STL10 experiment uses a 128-dimensional ball ($h = 128$).

We assess fairness and averaging distance in the latent space, denoted as $F_{\text{latent}}$ and $W_{\text{latent}}$, respectively. Additionally, we measure the reconstruction loss (RL) and the Wasserstein-2 distance between the prior and aggregated posterior distribution in the latent space, $W^2_{2,\text{latent}}$. Unlike the MNIST experiments, where the Wasserstein distance was used to measure metrics related in image space, we employ the FID score (Heusel et al., 2017) for CIFAR10 and STL10 due to its widespread use and reliability in measuring distances. Specifically, the F-metric and W-metric in the image domain and the gap between generated images and the dataset $W^2_{2,\text{image}}$ are calculated as:

$$F_{\text{images}} = \frac{2}{K(K-1)} \sum_{i=1}^{K-1} \sum_{j=i+1}^{K} \left| FID(\mu, \mu_i) - FID(\mu, \mu_j) \right|, \tag{17}$$

$$W_{\text{images}} = \frac{1}{K} \sum_{i=1}^{K} FID(\mu, \mu_i), \tag{18}$$

$$W^2_{2,\text{image}} = FID\left( \mu_0, \frac{1}{K} \sum_{k=1}^{K} \mu_k \right) \tag{19}$$

where $\mu$ is the empirical distribution of generated images, $\mu_1, \dots, \mu_K$ are the images for each label in the dataset, and $FID()$ is the FID score (Heusel et al., 2017). We report the quantitative results in Table 5 for the CIFAR10 experiment and Table 6 for the STL10 experiment.

The proposed methods outperform baselines across nearly all metrics. For CIFAR10, us-MFBSW and es-MFBSW deliver the best results, with us-MFBSW excelling in image domain metrics like $W^2_{2,\text{image}}$, $F_{\text{image}}$, and $W_{\text{image}}$. On STL10, es-MFBSW stands out, achieving the best $W^2_{2,\text{latent}}$, $W^2_{2,\text{image}}$, and $F_{\text{image}}$, while also improving fairness in the latent space with the lowest $F_{\text{latent}}$, while us-MFBSW does its best at reducing the averaging distance both in latent and image domain, which are $W_{\text{latent}}$ and $W_{\text{image}}$, respectively.

Overall, compared to the baselines, the proposed methods achieve greater geometric fairness and bring the generated images closer to the dataset distribution in both latent and image spaces, though this comes at the expense of reduced image reconstruction quality.

