# OpenReview forum: "Towards Marginal Fairness Sliced Wasserstein Barycenter"
_ICLR.cc/2025/Conference — ICLR 2025 Spotlight_

### Official Review · Reviewer_LqjG · 2024-11-02

**Soundness:** 3
**Presentation:** 3
**Contribution:** 3
**Rating:** 8
**Confidence:** 3

**Summary:**

This paper presents the marginal fairness sliced Wasserstein barycenter (MFSWB) problem. The goal of MFSWB is to achieve fairness of barycenter to marginals. To solve the problem, the Lagaragian format and three surrogate formulations are proposed, from biased estimation to unbiased estimation. Both time and space complexities are analyzed.

For the experiments, Gaussian toy data is utilized for verification. Then, point cloud, color hamonization, and SWAE experiments are conducted on more realistic data.
Both fairness metirc and barycenter metrics are reported.

**Strengths:**

- The presented problem is novel and reasonable. If fairness of barycenter can be achieved w.r.t. marginals, then varous future applicaiton can be conducted e.g. ensuring fairness of ML algorithms.
- The paper organization, problem definition and writing is good and easy to follow. If audience has some Sliced Wasserstein knowledge, then it will be easy to follow the idea.
- The Lagaragian formulation as well as three surrogates are proposed to deal with the Fairness barycenter estimation with unbiased gradient estimator.
The space and time complexity analysis show that the proposed e-MFSWB has good space and time complexity compared with conventional SWB.

**Weaknesses:**

- Most of the experiments conducted are simple or on simple dataset, such as Gaussian, Simple Color Hamonization of two images or point cloud of only two shapes, AE on simple MNIST dataset.
I am wondering the scability and real application of the proposed method. For example, if more than two (e.g. >5) images or point clouds are adopted, what might be the results?
Or are there more realistic potential appplications of the proposed method? If related work or potential applications are discussed, it would be better. For example, CelebA dataset results or Fairness generation [R1].
- USWB first appears in Figure 1 and used in several places. But it is not defined.
- I can understand the F-metric is significantly improved with the proposed solution. But for W-metric why it is also improved. Are there feasible explanation or insights? Can you provide an explanation or discussion on why the W-metric improves along with the F-metric? This would be valuableto better understand the proposed method.

[R1] Fair generative modeling via weak supervision.

–––––––––––––––
After Rebuttal:
The authors' rebuttal addressed those questions comprehensively, I would like to increase my overall score to 8.

**Questions:**

- What does USWB stand for in the paper?
- Can the method extended to settings with more than two images or point cloud?

---

> ### Author Response · Authors · 2024-11-17
> **Response to Reviewer LqjG**
>
> We sincerely appreciate the reviewer’s time and insightful comments. We would like to further develop the discussion as follows:
>
> **Q8**. Most of the experiments conducted are simple or on simple dataset, such as Gaussian, Simple Color Hamonization of two images or point cloud of only two shapes, AE on simple MNIST dataset.
> I am wondering the scability and real application of the proposed method. For example, if more than two (e.g. >5) images or point clouds are adopted, what might be the results?
> Or are there more realistic potential appplications of the proposed method? If related work or potential applications are discussed, it would be better. For example, CelebA dataset results or Fairness generation [R1].
>
> [R1] Fair generative modeling via weak supervision.
>
> **A8**. Thank you for the reference. We have added the reference to the paper. We decided not to use the CelebA dataset because converting its labels to the desired format requires additional effort. Due to time constraints, we plan to leave experiments with the CelebA dataset for future work. Instead, we opted for the STL10 dataset, which shares the same image dimensions as CelebA. To further investigate the scalability of the proposed method, we have recently experiment with CIFAR10 ($d = 32 \times 32 \times 3$) and STL10 ($d = 64 \times 64 \times 3$) for the Sliced Autoencoder experiments. Below are the tables summarizing CIFAR10 and STL10 experiments. Overall, compared to the baselines, the proposed methods achieve greater fairness notion and bring the generated images closer to the dataset distribution in both latent and image spaces, though this comes at the expense of reduced image reconstruction quality. It is worth noting that we use FID score to measure F-metric and W-metric on images space of CIFAR10 and STL10. We added the details of these experiments into the revised version of our paper (Table 5 and Table 6 in page 28).
>
> Table 1: Comparison of methods with $\kappa_2 =0.5$ on CIFAR10 after 500 epochs.
>
> | Methods   | $\text{RL}$ ($\downarrow$) | $\text{W}_{2,\text{latent}}^{2}$ ($\downarrow$) | $\text{W}_{2,\text{image}}^{2}$ ($\downarrow$) | $\text{F}_{\text{latent}}$ ($\downarrow$) | $\text{W}_{\text{latent}}$ ($\downarrow$) | $\text{F}_{\text{images}}$ ($\downarrow$) | $\text{W}_{\text{images}}$ ($\downarrow$) |
> |----------------------------|----------------------------|-------------------------------------------------|------------------------------------------------|------------------------------------------|--------------------------------------------|--------------------------------------------|--------------------------------------------|
> | SWAE   | 0.640  | 6.101   | 141.984| 0.280    | 4.585    | 46.006   | 178.798  |
> | UBSW | 0.640   | 6.104  | 135.944  | 0.228  | 4.572 | 44.024    | 174.322  |
> | MFSWB $\lambda = 0.1$   | 0.640  | 6.097    | 142.530  | 0.281  | 4.585   | 46.080  | 179.210 |
> | MFSWB $\lambda = 1.0$ | 0.641  | 6.092  | 142.289  | 0.279  | 4.578   | 46.076 | 179.135    |
> | MFSWB $\lambda = 10.0$   | 0.640 | 6.100   | 141.503 | 0.282 | 4.585   | 46.088   | 178.373   |
> | s-MFBSW  | 0.640  | 6.103   | 134.766 | 0.218  | 4.569     | 42.503| 173.530  |
> | us-MFBSW | 0.642 | 6.088  | **131.934**   | **0.209**| 4.546   | **39.329**    | **171.204**  |
> | es-MFBSW | 0.642  | **6.060**  | 132.170   | 0.212  | **4.534**  | 40.642   | 171.573
>
> Table 2: Comparison of methods with $\kappa_2 =0.5$ on STL10 after 500 epochs.
>
> | Methods   | $\text{RL}$ ($\downarrow$) | $\text{W}_{2,\text{latent}}^{2}$ ($\downarrow$) | $\text{W}_{2,\text{image}}^{2}$ ($\downarrow$) | $\text{F}_{\text{latent}}$ ($\downarrow$) | $\text{W}_{\text{latent}}$ ($\downarrow$) | $\text{F}_{\text{images}}$ ($\downarrow$) | $\text{W}_{\text{images}}$ ($\downarrow$) |
> |--|----|-----|---|----|-----|----|----|
> | SWAE     | 0.613   | 16.826    | 301.397    | 0.647    | 15.699    | 25.827      | 199.175   |
> | UBSW   | 0.616    | 16.908   | 301.143    | 0.585 | 15.719   | 24.905    | 199.918   |
> | MFSWB $\lambda = 0.1$      | 0.614   | 16.823   | 301.704    | 0.647      | 15.698   | 25.637   | 199.662   |
> | MFSWB $\lambda = 1.0$      | 0.614  | 16.814   | 301.505    | 0.647   | 15.688   | 25.790   | 199.307  |
> | MFSWB $\lambda = 10.0$     | 0.613    | 16.831   | 301.370  | 0.648   | 15.705 | 25.546   | 199.168    |
> | s-MFBSW  | 0.613   | 16.842    | 302.632   | 0.580   | 15.658   | 23.520  | 200.262   |
> | us-MFBSW  | 0.616 | 16.830   | 297.952   | 0.586   | **15.645**  | 23.638   | **197.057** |
> | es-MFBSW  | 0.616  | **16.796** | **296.548**   | **0.557**  | 15.658  | **22.551**  | 199.117  |

---

> ### Author Response · Authors · 2024-11-17
> **Response to Reviewer LqjG part 2**
>
> **Q9**. USWB first appears in Figure 1 and used in several places. But it is not defined.
>
> **A9**. Thank you for pointing out, USWB stands for Uniform Sliced Wasserstein Barycenter, which sets marginal weights in Sliced Wasserstein Barycenter problem to be uniform. We added the explanation to the revision.
>
> $\min_{\mu}\text{USWB}(\mu; \mu_{1:K});
>     \quad \text{USWB}(\mu;\mu_{1:K})= \frac{1}{K} \sum_{k=1}^K \text{SW}_p^p (\mu,\mu_k).$
>
> **Q10**. I can understand the F-metric is significantly improved with the proposed solution. But for W-metric why it is also improved. Are there feasible explanation or insights? Can you provide an explanation or discussion on why the W-metric improves along with the F-metric? This would be valuable to better understand the proposed method.
>
> **A10**. Thank you for your insightful question. It is worth noting that MFSWB has higher W-metric than USWB in the Gaussian cases in Section 4.1. For other applications, MFSWB does achieve lower USWB. The reason for this phenomenon is due to the chosen optimization procedure and the different landscapes of MFSWB and USWB. Therefore, the obtained barycenters might be only local optimums. At those optimums, MFSWB is better than USWB in both F-metric and W-metric. Nevertheless, we project that MFSWB would have lower F-metric than USWB and have higher W-metric than USWB at the global optimum. We would like to refer to the simple simulation of Gaussians in Section 4.1 as the ideal example. We will leave the investigation about optimization to future works.
>
> **Q11**. What does USWB stand for in the paper?
>
> **A11**. Thank you for your question, as mentioned in question 9, USWB stands for Uniform Sliced Wasserstein Barycenter i.e., the Sliced Wasserstein Barycenter problem where all marginals have equal weights. We added the explanation to the revision.
>
> **Q12**. Can the method extended to settings with more than two images or point cloud?
>
> **A12**. Thank you for your questions. The barycenter problem is defined with an arbitrary number of marginals i.e., $K>1$. Similarly, our proposed MFSWB is  defined with $K>1$ marginals e.g., $K=2,3,...$. In Section 4.1., we consider $K=4$ marginals of Gaussians. In Section 4.4, we consider $10$ marginals where each marginal is a class distribution. The main reason we used two images and two point clouds in Section 4.2 and 4.3 is that averaging more than two images and point clouds might lead to unnatural color palettes and 3D shapes for human's perception due to the heterogeneity of all marginals.  For example, an average of many 3D shapes of car can be just a 3D ball. Therefore, it requires more efforts to select marginals to obtain a visually interpretable barycenter as a 3D shape prototype. Since the paper focuses on the concept of MFSWB, we leave the careful investigation on the effect of varying the number of marginals to future works.
>
> Please feel free to ask if you still have any other questions.

---

> ### Comment · Reviewer_LqjG · 2024-11-25
> **Thanks for the rebuttal**
>
> The authors' rebuttal comprehensively addressed my questions and concerns, with additional experimental results.
> I would like to increase my score from 6 to 8.

---

> > ### Author Response · Authors · 2024-11-25
> > **Response to  Reviewer LqjG**
> >
> > We would like to thank the reviewer for increasing the score to 8. We will continue revising our paper based on constructive feedback from the reviewer and other reviewers.  Please feel free to ask if you still have other questions.
> >
> > Best regards,
> >
> > Authors,

---

### Official Review · Reviewer_6MLn · 2024-11-02

**Soundness:** 3
**Presentation:** 3
**Contribution:** 3
**Rating:** 6
**Confidence:** 4

**Summary:**

This paper aims at achieving marginal fairness sliced wasserstein barycenter (MFSWB). To tackle the problem, the authors define the MFSWB as a constraint SWB problem. Due to the computational disadvantages of the formal definition, the authors propose two hyperparameter-free and computationally tractable surrogate MFSWB problems that implicitly minimize the distances to marginals and encourage marginal fairness at the same time. To further improve the efficiency, the authors perform slicing distribution selection and obtain the third surrogate definition by introducing a new slicing distribution that focuses more on marginally unfair projecting directions. The experimental results demonstrate the favorable performance of the proposed MFSWB performance.

**Strengths:**

1. The paper is well-organized and the writing is good.
2. The paper has solid theoretical proofs.
3. The story line is easy to follow.

**Weaknesses:**

Please refer to questions.

**Questions:**

1. Can the authors elaborate on the significance of MFSWB? The description in current version is somewhat high-level.
2. Can the authors provide a clearer description about the formulation in line 148?
3. What does the symbol $\sharp$ in Eq.(1) denote?
4. What does the $x_i(y_i)$ in line 144 denote?
5. What does the $\pi$ in line 144 denote?
6. What modification to the methodology presented in this paper are necessary to replace SWD with GSWD [1] or ASWD [2]?

[1] Kolouri, Soheil, et al. "Generalized sliced wasserstein distances." Advances in neural information processing systems 32 (2019).

[2] Chen, Xiongjie, Yongxin Yang, and Yunpeng Li. "Augmented sliced Wasserstein distances." arXiv preprint arXiv:2006.08812 (2020).

---

> ### Author Response · Authors · 2024-11-17
> **Response to Reviewer 6MLn**
>
> We genuinely appreciate the reviewer's time and thoughtful feedback. We would like to elaborate on the discussion as follows:
>
> **Q2**. Can the authors elaborate on the significance of MFSWB? The description in the current version is somewhat high-level.
>
> **A2**.Thank you for your question on the impact of the MFSWB. We would like to elaborate on the significance of MFSWB as follow.
> MFSWB generalizes the notion of a center to the space of probability measures. In particular, the conventional center of a set of points in a vector space corresponds to the center of the smallest circle enclosing all the points in the set. Similarly, MFSWB can be seen as the center of a set of points in the space of probability measures, defined using the SW distance geometry. This generalized notion of a center enables a broader application, such as generalizing k-center clustering to handle probability measures instead of vectors. Furthermore, as demonstrated in the paper, MFSWB can identify the center of 3D shapes (e.g., human brains, 3D objects), which can serve as prototypes for future tasks. For instance, the 3D prototype can be used to generate new 3D shapes via a deformation function. Finally, MFSWB can facilitate learning fair representations across classes by ensuring that the regions of high probability for each class distribution are approximately equal. This fair representation can implicitly enhance fairness in downstream tasks.
>
>
> **Q3**. What does the symbol $\sharp$ in Eq.(1) denote?
>
> **A3**. Thank you for pointing out. The symbol $\sharp$ denotes the pushforward operator. In particular, $\theta\sharp \mu$ denotes the pushfoward measure of $\mu$ through the function $f(x)=\theta^\top x$. We have added an explanation to the revision.
>
> **Q4**. Can the authors provide a clearer description about the formulation in line 148?
>
> **A4**In the formula, $[n]$ denotes the  set $\{1,2,\ldots,n\}$, $\sigma_1:[n] \to [n]$ is the permutation function such that $x_{\sigma_1(1)} \leq x_{\sigma_1(2)} \leq \ldots \leq x_{\sigma_1(n)}$ or $x_{\sigma_1(1)} \geq x_{\sigma_1(2)} \geq \ldots \geq x_{\sigma_1(n)}$. Similarly, $\sigma_2:[n] \to [n]]$ is the permutation function such that $y_{\sigma_2(1)} \leq y_{\sigma_2(2)} \leq \ldots \leq y_{\sigma_2(n)}$ or $y_{\sigma_2(1)} \geq y_{\sigma_2(2)} \geq \ldots \geq y_{\sigma_2(n)}$, and  $\sigma_2^{-1}$ is the argsort operator. The optimal transport map is constructed as $\sigma = \sigma_1 \circ \sigma_2^{-1}$. We added this to the revision of the paper.
>
> **Q5**. What does the $x_i(y_i)$ in line 144 denote?
>
> **A5**. In that line, $\{x_1,\ldots,x_n\}$ and $\{y_1,\ldots,y_n\}$ are the supports sets of two empirical measures $\mu =1/n\sum_{i=1}^n \delta_{x_i}$ and $\nu=1/n\sum_{i=1}^n \delta_{y_i}$
>
> **Q6**. What does the $\pi$ in line 144 denote?
>
> **A6**. We assume that you are asking for the notion $\phi$ in line 144, $\phi$ is the parameter of the barycenter as defined in line 128. In the fixed support barycenter case, $\phi$ is the weights of supports. In the free support barycenter case, $\phi$ is the supports.
>
> **Q7**. What modifications to the methodology presented in this paper are necessary to replace SWD with GSWD [1] or ASWD [2]?
>
> [1] Kolouri, Soheil, et al. "Generalized sliced wassertein distances." Advances in neural information processing systems 32 (2019).
>
> [2] Chen, Xionglie, Yongxin Yang, and Yunpeng Li. "Augmented sliced Wassertein distances." arXiv preprint arXiv:2006.08812 (2020).
>
> **A7**. Thank you for your insightful questions and your references. We added these references to the revision. GSWD and ASWD utilize different types of projections compared to SW. In particular, GSWD uses non-linear projections such as circular, polynomial, and so on, while ASWD utilizes an injective non-linear augmentation function followed by a linear projection. Replacing SWD by GSWD and ASWD results in different marginal fairness barycenter problems. This change leads to a geometric change in the barycenter, which is worth a careful investigation in our opinion. Moreover, the notion of a marginal fairness barycenter can adapt to any metric, such as Wasserstein, SWD, GSWD, and ASWD. With each choice of metric, an implied geometry is used to seek the barycenter. Since the marginal fairness barycenter is the main focus of the paper, we leave the investigation with other metrics to future works.
>
> Please feel free to ask if you still have any other questions.

---

> > ### Comment · Reviewer_6MLn · 2024-11-25
> > **Response to authors**
> >
> > We would like to thank the reviewer for the time and constructive feedback. My concers are addressed, and I would keep my positive score.

---

> > > ### Author Response · Authors · 2024-11-25
> > > **Response to Reviewer 6MLn**
> > >
> > > We would like to thank the reviewer for your constructive reviews.  Please feel free to ask if you still have other questions.
> > >
> > > Best regards,
> > >
> > > Authors,

---

### Official Review · Reviewer_1X1Z · 2024-11-02

**Soundness:** 3
**Presentation:** 3
**Contribution:** 3
**Rating:** 8
**Confidence:** 3

**Summary:**

The traditional Sliced Wasserstein Barycenter (SWB) employs uniform weights between the barycenter and its marginals, leading to imbalances in the distances between them. This paper introduces the Marginal Fairness Sliced Wasserstein Barycenter (MFSWB), which aims to equalize these distances, ensuring a fairer representation. The authors propose three algorithms to address the MFSWB formulation and conduct experiments on synthetic Gaussian distributions, 3D point clouds, color harmonization, and generative models. The results demonstrate that the proposed method is highly effective, validating the authors' claims.

**Strengths:**

This paper addresses an important problem that has been overlooked. In traditional SWB, the weights between barycenter and marginals are uniform. However, this uniform weights actually leads to non-uniform distances between the barycenter and the marginals. The authors have a simple experiment demonstrating that in the paper. To ensure equal distances, they propose the MFSWB to address this problem. They define MFSWB such that the average distance to be within an $\epsilon$.

The proposed MFSWB is hard to solve. The authors proposed three surrogate definitions for MFSWB. The first one, called s-MFSWB, aims to minimize the maximal distance from the barycenter to the marginals. Yet, the s-MFSWB is biased. To rectify this, they proposed the second unbiased surrogate, us-MFSWB. They show that us-MFSWB's objective is an upper bound of s-MFSWB, meaning that minimizing us-MFSWB implicitly minimizes s-MFSWB. Furthermore, They also show that the approximation error of the gradient estimator of us-MFSWB reduces at the order of $\mathcal{O}(L^{−1/2})$. Since s-MFSWB and us-MFSWB both use the uniform distribution as the slicing distribution, which can empirically result in non-optimal in statistical estimation, the authors proposed the energy-based surrogate MFSWB (es-MFSWB) which considers the importance of the sampling directions.

The authors conducted extensive experiments evaluating the proposed methods and comparing the methods against the conventional SWB. The results are convincing. Using the synthetic Gaussian data, they showed that SWB produced the barycenter with non-uniform distances from the marginals, whereas the proposed methods yielded barycenters with approximately equal distances from the marginals. The authors also performed experiments on the 3D points clouds, Color Harmonization, and generative models with both quantitative results and qualitative results showing the effectiveness of the proposed method. The proposed es-MFSWB appears to be the best among all three proposed methods.

The authors shows the potential of using the proposed method in learning conditional generative models as an inverse barycenter problem.

**Weaknesses:**

I didn't really catch any weakness of this paper. The paper is complete with clear motivation, effective solutions and sufficient experimental results.

**Questions:**

Did you have a typo in line 506? Should $P_{f(X_k)}$ be equal to $\frac{1}{M} \sum_{i=1}^M \delta_{f(X_{ki})}$?

---

> ### Author Response · Authors · 2024-11-17
> **Response to Reviewer 1X1Z**
>
> We would like to thank the reviewer for the time and constructive feedback. We would like to extend the discussion as follow.
>
> **Q1**. Did you have a typo in line 506? Should $P_{f(X_k)}(x)$ be equal to $\frac{1}{M}\sum_{i=1}^M \delta_{f(X_{k,i})}$?
>
> **A1**. Thank you very much for pointing out the typo. It should be $\frac{1}{M}\sum_{i=1}^M \delta_{f_\phi(X_{k,i})}$. We fixed all related typos in the revision.
>
> Please feel free to ask if you still have any other questions.

---

> > ### Comment · Reviewer_1X1Z · 2024-11-26
> > **Response**
> >
> > Thank the authors for the clarification! I don't have other questions.

---

> > > ### Author Response · Authors · 2024-11-26
> > >
> > > Thank you very much for your constructive feedback. We will revise the paper based on the reviews from you and other reviewers.
> > >
> > > Best regards,
> > >
> > > Authors

---

### Author Response · Authors · 2024-11-18
**Summary**

Dear chairs and reviewers,

First, we would like to thank the reviewers for spending time reviewing our paper and providing constructive feedback. We would like to summarize some changes and additional results as follows:

1. As some reviewers point out, there are some typos in the paper, we have fixed them in the revision of the paper.

2. As suggested by **reviewer LqjG**, we have extended our experiments to include the CIFAR10 and STL10 datasets for the Sliced Wasserstein Autoencoder, aiming to assess the proposed method's scalability. The results demonstrate that the proposed marginal fairness sliced Wasserstein barycenter approaches consistently show their benefits in learning fair representation with scalability.

3. For reviewers' questions, we extend the discussion to corresponding threads. We revised the revision based on the reviewers' suggestions. Changes are made in blue in the revision.

Best regards,

Authors

---

### Meta-Review · Area_Chair_kyje · 2024-12-19

**Metareview:**

The traditional Sliced Wasserstein Barycenter (SWB) employs uniform weights between the barycenter and its marginals, often resulting in imbalances in the distances between them. This paper introduces the Marginal Fairness Sliced Wasserstein Barycenter (MFSWB), designed to equalize these distances and ensure a more balanced representation. To address the MFSWB formulation, the authors propose three algorithms and evaluate their effectiveness through experiments on synthetic Gaussian distributions, 3D point clouds, color harmonization, and generative models. The results demonstrate the high efficacy of the proposed method, providing strong validation for the authors' claims.  The paper is well-written and well-organized with sufficient theoretical and empirical justification, making a valuable contribution to the community.  The rebuttal addressed the concerns raised by the reviewers in their initial assessments. This led to a consensus among all three reviewers, who agreed to accept the paper based on its innovation and empirical contributions. The AC concurs with the reviewers and recommends accepting the paper.

**Additional Comments On Reviewer Discussion:**

Reviewer 1X1Z did not raise significant concerns about the paper.

Reviewer 6MLn asked several clarification questions, which the authors effectively addressed in their rebuttal.

In the initial review, Reviewer LqjG expressed concerns about the scalability and real-world applicability of the proposed method, as well as the model being tested on relatively simple applications. In response, the authors provided comprehensive feedback and additional results, fully addressing these concerns. As a result, Reviewer LqjG increased the score from 6 to 8.

After the rebuttal, all reviewers reached a consensus to accept the paper.

---

### Decision · Program_Chairs · 2025-01-22

Accept (Spotlight)